# Unveiling LLMs: The Evolution of Latent Representations in a Dynamic Knowledge Graph

Marco Bronzini[*1, 2], Carlo Nicolini[2], Bruno Lepri[2, 3], Jacopo Staiano[1], and Andrea Passerini[1]

[1]University of Trento, Trento, Italy
[2]Ipazia S.p.A., Milan, Italy
[3]Fondazione Bruno Kessler (FBK), Trento, Italy

## Abstract

Large Language Models (LLMs) demonstrate an impressive capacity to recall a vast range of factual knowledge. However, understanding their underlying reasoning and internal mechanisms in exploiting this knowledge remains a key research area. This work unveils the factual information an LLM represents internally for sentence-level claim verification. We propose an end-to-end framework to decode factual knowledge embedded in token representations from a vector space to a set of ground predicates, showing its layer-wise evolution using a dynamic knowledge graph. Our framework employs activation patching, a vector-level technique that alters a token representation during inference, to extract encoded knowledge. Accordingly, we neither rely on training nor external models. Using factual and common-sense claims from two claim verification datasets, we showcase interpretability analyses at local and global levels. The local analysis highlights entity centrality in LLM reasoning, from claim-related information and multi-hop reasoning to representation errors causing erroneous evaluation. On the other hand, the global reveals trends in the underlying evolution, such as word-based knowledge evolving into claim-related facts. By interpreting semantics from LLM latent representations and enabling graph-related analyses, this work enhances the understanding of the factual knowledge resolution process.

## 1 Introduction

Several studies have demonstrated the ability of Large Language Models (LLMs) to store and recall an impressive variety of common-sense and factual knowledge (Meng et al., 2022; Jiang et al., 2020; Shin et al., 2020; Brown et al., 2020; Petroni et al., 2019). However, investigating how LLMs leverage this knowledge and their reasoning remains an ongoing research challenge. This work studies LLMs' knowledge resolution mechanism and represents its underlying evolution as a dynamic knowledge graph. It addresses three research questions: (i) Which factual knowledge do LLMs use to assess input truthfulness? (ii) How does this knowledge evolve across layers? (iii) Are there any distinctive patterns in its evolution?

We investigate how factual knowledge, encoded in the latent spaces of LLMs, changes during inference when tasked with claim verification. Specifically, we propose a framework[1] to reveal the factual information an LLM holds internally when evaluating the truthfulness of short claims such as *"Charlemagne was crowned emperor on Christmas Day"*. It unveils non-trivial insights into the internal workings of LLMs as exhibited in Figure 1. Analysing the vector space of LLMs, also known as latent representations, implies tracking the evolution of token representations across the model's hidden layers (Vaswani et al., 2017) and segmenting inference into discrete time steps: layer $t$ at time $t$, layer $t + 1$ at time $t + 1$, and so forth.

---

[*]Corresponding author. marco.bronzini-1@unitn.it
[1]The framework and its code are available as a Python package named `Latent-Explorer`.

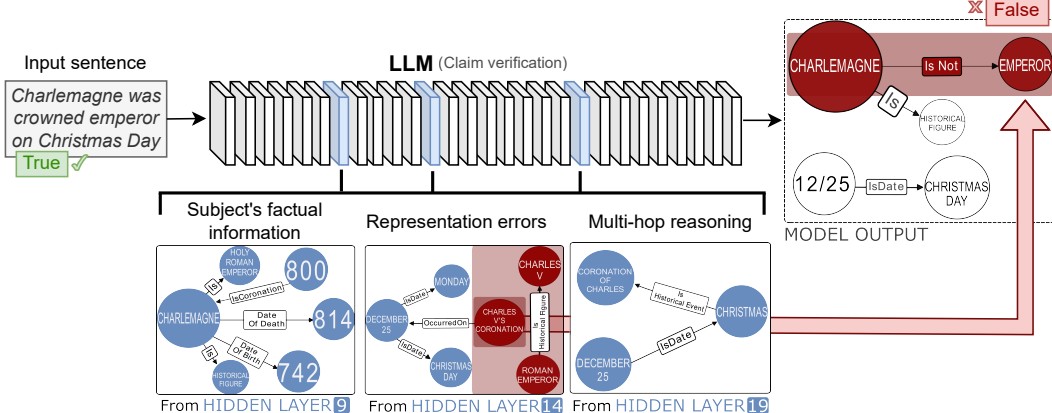

Figure 1: Illustrative insights from unveiling the process of factual knowledge resolution within an LLM using the proposed patching-based framework.

Here, we develop a framework (Figure 2) to jointly decode factual knowledge embedded in LLMs' latent representations and represent its dynamics using a graph representation. Initially, we elicit a model's behaviour by assigning a language model (LLaMa2; Touvron et al., 2023) the task of verifying entire claims, while storing token representations during inference. Next, we prompt a separate model inference (Figure 2) to decode the semantics embedded in these representations using activation patching (Zhang & Nanda, 2023). Since patching is a token-level technique, we beforehand define a function that maps the input claim's token representations to a vector representation. This matrix-to-vector mapping function creates a summarised and condensed representation of the input using a weighted sum, exploiting the additive property of token representations to combine their embedded semantics. It offers an alternative to multi-token patching, which treats each token independently and extends single-token patching (Ghandeharioun et al., 2024). Afterwards, the inference patched with the input's summary interprets its encoded semantics as ground predicates such as IsDate(Christmas Day, December 25). This formalisation has two advantages: (i) formatting factual knowledge consistently for subsequent knowledge graph generation and (ii) connecting information using logical symbols (∧ and ¬). Lastly, we represent the extracted knowledge, and its evolution, as a dynamic knowledge graph (Figure 2), combining a multi-relational graph of entities and relations (Wang et al., 2017) with a dynamic graph[2] (Harary & Gupta, 1997). This allows us to visually represent the process of factual knowledge resolution using the model's layers as the graph's temporal dimension.

After collecting factual and common-sense claims from two well-known claim verification datasets, FEVER (Thorne et al., 2018) and CLIMATE-FEVER (Diggelmann et al., 2020), we showcase two use cases for the outputs of the proposed framework: local and global interpretability analyses on the factual knowledge decoded from latent representations. Local interpretability highlights knowledge centrality in LLM reasoning: from the subject's factual information and multi-hop reasoning to representation errors causing erroneous evaluations (see Figure 1). On the other hand, the graph representation helps reveal global trends in LLMs' factual knowledge resolution process, from middle-layer importance to word-based information evolving into claim-related facts. Our main contribution is an end-to-end framework that jointly accomplishes several tasks:

- decoding the semantics embedded in the latent space of LLMs, in the form of ground predicates; without relying on external models or training processes;
- extending single-token patching by exploiting the additive property of LLM's token representations to probe the semantics of multiple tokens jointly;
- representing the encoded factual knowledge and tracing its underlying evolution using a graph representation;

---

[2]Where nodes (entities) and edges (relations) change over time.

- enabling interpretability analyses at both global and local levels, revealing, for instance, word-based knowledge evolving into claim-related facts and representation errors that cause incorrect evaluations.

By decoding semantics from LLM latent representations and enabling graph-related analyses, this framework advances our understanding of the factual knowledge resolution process and the mechanistic interpretability of language models.

The rest of the work is organized as follows. Section 2 reviews related literature, Section 3 details our approach.This is followed by Section 4 outlining our experiments, and Section 5 that presents concluding remarks.

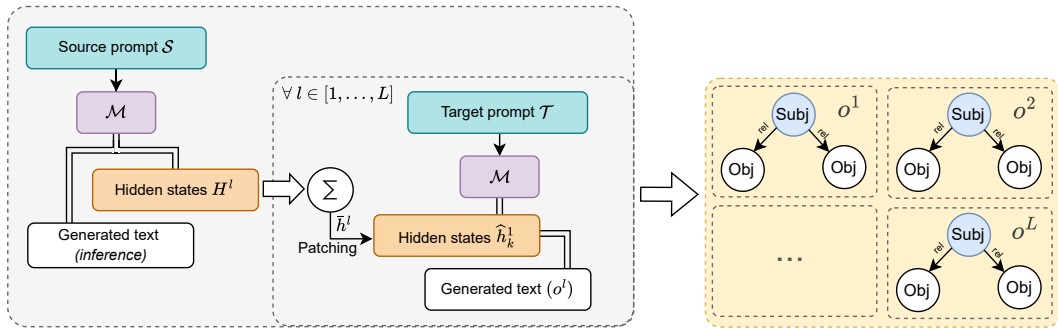

Figure 2: The patching-based framework decodes the factual knowledge from LLM latent representations. The outputs are represented in a dynamic knowledge graph.

## 2 Related Work

Activation patching is traditionally applied in mechanistic interpretability to study components and internal workings of machine-learned models. Conventional workflows involve eliciting a behaviour to observe, discovering patterns via activation patching and generating functional hypotheses (Conmy et al., 2024).

Specifically, the patching technique studies the model's computation by altering its latent representations, the token embeddings in transformer-based language models, during the inference process (Ghandeharioun et al., 2024). For example, Meng et al. (2022) focused on causal tracing by replacing the latent representation of a noise-corrupted inference with the correct ones, identifying the crucial activations in rectifying the correction. Ghandeharioun et al. (2024) proposed an LLM-based framework for semantically inspecting the latent representations of LLMs. The authors studied the entity resolution process across the early layers of LLMs by patching the latent representation of a subject entity (e.g., "Alexander the Great") into a separate inference tasked to describe it. Alternatively, Yang et al. (2024) studied LLMs' multi-hop reasoning via a framework projecting latent representations to vocabulary space. They examined LLMs' handling of first-hop and second-hop reasoning tasks in completing two-hop factual propositions, assessing latent bridging entity representation for connecting information fragments. Teehan et al. (2024) proposed instead a framework to generate high-quality latent representations for new concepts using a small number of example sentences or definitions. These studies, along with others conducted recently (Mallen et al., 2023; Hernandez et al., 2023; Jiang et al., 2020) have investigated the factual understanding of LLMs regarding single entities in scenarios involving knowledge completion, such as incomplete triplets. On the contrary, our research focuses on entire sentences, exploring the extensiveness and evolution of factual knowledge embedded within LLMs when tasked to evaluate the input truthfulness. This represents a step towards understanding the LLMs' factual knowledge resolution process rather than factual knowledge of single entities.

## 3  Methodology

Following the framework proposed by Ghandeharioun et al. (2024), we leverage the activation patching technique to decode the semantics, in the form of factual information, embedded within an LLM vector space and represent its evolution across the model's layers using a dynamic knowledge graph. The procedure is outlined in Figure 2.

Given a language model $\mathcal{M}$ with $L$ hidden layers, and a prompt $\mathcal{S}$ containing an input sentence, we probe the tokens' latent representations[3] obtained at each hidden layer $l \in [1, ..., L]$ during the inference of $\mathcal{M}$ on $\mathcal{S}$ (residual stream). We execute a separate inference of the model $\mathcal{M}$ with a different prompt $\mathcal{T}$ to decode the semantics embedded in these representations. This prompt $\mathcal{T}$ contains a special placeholder token "$x$" for the patching process: substituting its latent representation with a summary of the input's latent representations from the original inference. We specifically intervene during the model computation on prompt $\mathcal{T}$ by mapping the placeholder token's embedding with a weighted sum of the token embeddings of the input sentence obtained at the $l$-th hidden layer of the model inference on $\mathcal{S}$. The execution of this patched inference generates then a structured text $o^l$, containing a list of factual information. The procedure is repeated for the different values of $l \in [1, \ldots, L]$. All the generated texts are then represented as a dynamic knowledge graph, with $l$ being the graph's dynamic and temporal dimension.

Essentially, the proposed framework converts the LLM internal vector space into a human-understandable semantic graph by collectively probing the encoded semantics of multiple token representations. The following further details the different procedure steps.

### 3.1  Prompt Definition

Initially, we define a template for the model instruction, the source prompt $\mathcal{S}$, encompassing three different semantic parts: (i) a *system instruction* describing how to accomplish the claim verification task, (ii) an *input-output example* to help the model generate the desired output, and (iii) the *input sentence* (Figure 3). The desired output is a structured text with two attributes: (i) a binary label indicating the truthfulness of the sentence, and (ii) a list of facts supporting such evaluation. The facts are represented as a conjunction of ground literals: asserted or negated predicates. For instance, the factual information necessary to evaluate the input claim "Edgar Allan Poe wrote Hamlet" can be represented as: `AuthorOf(Hamlet, William Shakespeare)` $\land$ `¬AuthorOf(Hamlet, Edgar Allan Poe)`, where $\land$ and $\neg$ indicate the logical conjunction and negation respectively. Using ground literals has two advantages: (i) formatting factual knowledge consistently for the subsequent knowledge graph generation, and (ii) stimulating the language model to associate and contrast factual information using its logic symbolic knowledge (De Smet et al., 2023). Our preliminary experiments indicated that using a simple subject-predicate-object (SPO) triple representation leads to sub-optimal outcomes, resulting in more isolated and subject-focused

---

[3]The terms *latent/vector representations/space*, *embeddings*, and *hidden states* are used interchangeably.

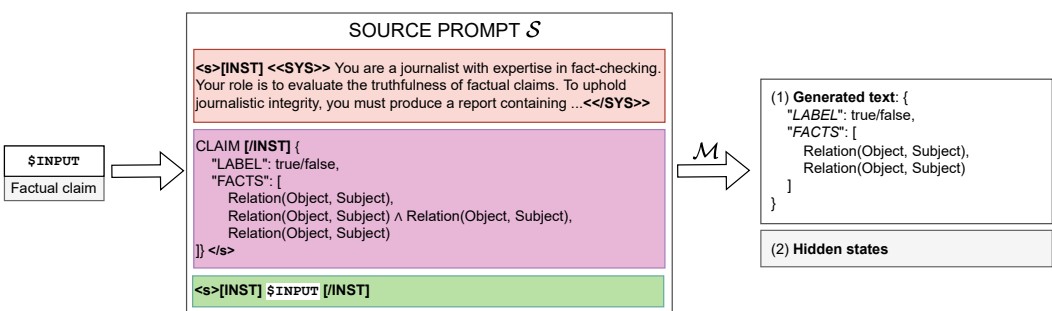

Figure 3: Inference of $\mathcal{M}$ on the source prompt $\mathcal{S}$.

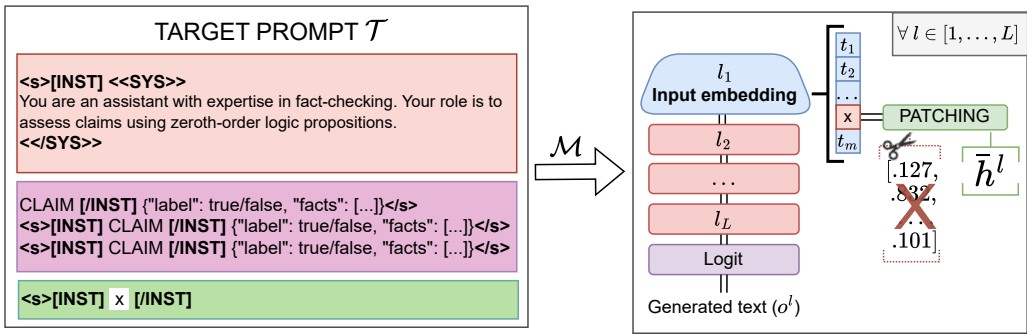

Figure 4: Patching operation during the inference of model $\mathcal{M}$ on the target prompt $\mathcal{T}$.

information. The full prompt is shown in Appendix A. We then apply this template to a given input sentence and run the language model $\mathcal{M}$ on $\mathcal{S}$, producing a structured output and storing the hidden states ($H^l$), the token representations across the model's layers.

## 3.2  Patching

We denote the matrix $H^l$ as the latent representations obtained from the $l$-th hidden layer of the model's inference on the source prompt $\mathcal{S}$. We then consider a separate inference of the model $\mathcal{M}$ on a different prompt, a target sequence of $m$ tokens $\mathcal{T} = \langle t_1, ..., t_m \rangle$. This prompt mimics the source prompt to generate similar outputs but serves as a propositional probe. It decodes the semantics within the latent representations via activation patching. To perform the patching operation, we include a placeholder token, the character $"x"$, within this prompt. We also augment the in-context examples (1 to 3) and reduce the system instruction to boost the model's in-context abilities during inference. The full prompt is shown in Appendix B. This patching operation consists of replacing the vector representation of the placeholder token ($\widehat{h}_k^1$, where $k$ is the position of $x$ in $\mathcal{T}$) with a summary of the vector representations of the input sentence from original inference ($\mathcal{M}$ on $\mathcal{S}$), leaving the other latent representations unchanged, and letting $\mathcal{M}$ proceed with the inference (Figure 4). Since activation patching is a vector-level technique, we formally define a matrix-to-vector mapping function $f(H^l, W) : \mathbb{R}^{n \times d} \mapsto \mathbb{R}^d$, parameterized by $W$, that computes a weighted sum of the latent representations of the input part $\mathcal{I} \subset \mathcal{S}$ of the source prompt:

$$f(H^l, W) := \sum_{i=1}^{L^I} w_i h_i^l = \bar{h}^l \tag{1}$$

where the input part refers to the input sentence, the factual claim, and $L^I = |\mathcal{I}|$ is its length. We set the weights $W$ of the weighted sum by performing part-of-speech tagging[4] on the input claim. Nouns and verbs are assigned a weight equal to zero to all but their end token which receives a weight of one (Figure 5), emphasising sentence's entities and predicates in this summary vector representation. We focus on end tokens based on Meng et al.'s (2022) study, which found that the model forms a subject representation at the final token of an entity name. Appendix G shows that including all tokens, especially stop words, is detrimental. Additionally, empirical experiments showed that using single tokens leads to meaningless texts for punctuation and single-word information for the last input token.

Patching is then applied by replacing $\widehat{h}_k^1$ with $\bar{h}^l$ in the model's input embedding layer, as visually exhibited in Figure 4. This model inference, patched with the summary input representation from $l$-th layer, generates a structured text $o^l$, structurally equal to the one from the original model inference ($\mathcal{M}$ on $\mathcal{S}$). By applying this procedure for all values of $l \in [1, \ldots, L]$, we produce structured outputs using all latent representations of $\mathcal{M}$.

---

[4]https://spacy.io/models/en

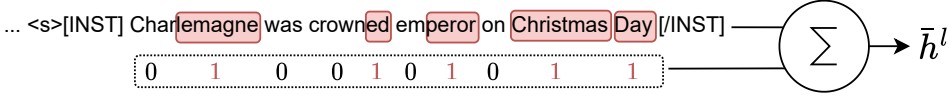

Figure 5: Example of the binary weights for the input's tokens $\mathcal{I} \subset \mathcal{S}$. These weights are then used to combine the tokens' vector representations via a weighted sum.

### 3.3 Knowledge Graph Generation

We use a knowledge graph to represent the list of ground literals, the factual information, included in the structured output (Figure 3) generated by each patched inference ($o^l \in \mathcal{O} \mid \mathcal{M}$ on $\mathcal{T}$) and the original inference ($\mathcal{M}$ on $\mathcal{S}$). We first turn literals into subject-predicate-object (SPO) triples using simple rewriting rules:

$$
\begin{aligned}
(\neg)Relation(object_1, object_2) &\rightarrow \langle object_1, (not)relation, object_2 \rangle \quad \text{for binary predicates} \\
(\neg)isProperty(object) &\rightarrow \langle object, is(not), property \rangle \quad\quad\quad \text{for unary predicates}
\end{aligned}
\tag{2}
$$

Afterwards, we represent all the triples, yielded from $o^l$, as a knowledge graph (Figure 6). We eventually concatenate all the graphs generated for the different values of $l$, creating a dynamic graph that exhibits the factual knowledge evolution across the model's layers.

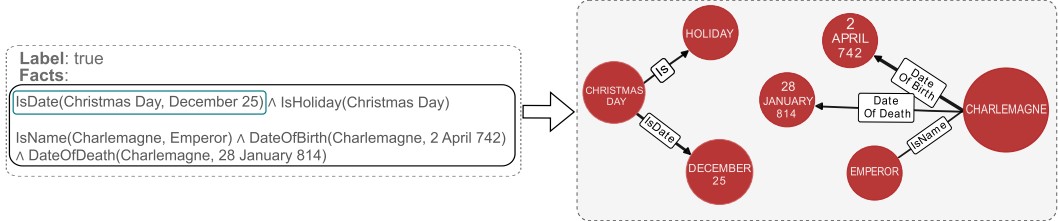

Figure 6: Graph generation process from the ground literals to a knowledge graph. This process is performed for each $o^l \in \mathcal{O}$ and the inference output.

## 4 Experiments

This section showcases our framework on two experimental use cases: local and global interpretability analyses of the extracted factual knowledge. After describing our experimental setup in Section 4.1 and evaluating the effectiveness of the language model in Section 4.2, we present two interpretability analyses, a local analysis of the factual information of three distinct input claims in Section 4.3, and global analysis of the patterns in the factual knowledge resolution process in Section 4.4.

### 4.1 Experimental Setup

To analyse the model behaviour in recalling factual knowledge to support or debunk input sentences, we prompt a language model with a collection of factual and common-sense claims sampled from two well-known claim verification datasets: *FEVER* (Thorne et al., 2018) and *CLIMATE-FEVER* (Diggelmann et al., 2020). As in conventional mechanistic interpretability workflows (Conmy et al., 2024), these inputs serve to elicit a model behaviour, unfolding the factual information an LLM holds internally (Section 1).

Each claim is labelled as Supported (True), Refuted (False) or NotEnoughInfo. For instance, the claim *"Charlemagne was crowned emperor on Christmas Day"* from the FEVER dataset is classified as Supported, whereas *"Berlin has a population of 4 million people"* is classified as Refuted. These are all factual claims and a language model should rely on specific factual

knowledge to evaluate their truthfulness. On the other hand, *CLIMATE-FEVER*, a dataset mimicking the FEVER methodology, compounds to real-world claims regarding climate change (Diggelmann et al., 2020). Its claims may require more common-sense reasoning, and elicit subjective dichotomies. For example, the claim *"Global warming is increasing the magnitude and frequency of droughts and floods"* is labelled as `Supported`, whereas *"Renewable energy investment kills jobs"* is classified as `Refuted`.

We initially filter the two datasets by (i) excluding the claims labelled with *NotEnoughInfo*, avoiding prompting the model with unverifiable sentences, and (ii) considering claims neither too short nor too long ($35 \leq |characters| \leq 120$). Afterwards, we randomly sample 1,000 claims from FEVER (*Supported*: 726, *Refuted*: 274) and 400 from CLIMATE-FEVER (*Supported*: 274, *Refuted*: 126). We use the LLaMA 2 language model (Touvron et al., 2023) in its 7-billion instruction-tuned version[5] to showcase our framework. Appendix F compares the output generated by similar language models: the model's 13-billion version, the newest LLaMA 3 (AI@Meta, 2024) and the 7-billion Falcon model (Almazrouei et al., 2023).

### 4.2 Classification Performance

**Method.** We here assess the effectiveness of the language model $\mathcal{M}$ to classify the input sentences. This helps to better understand whether the desired model behaviour is properly elicited: recalling helpful knowledge to evaluate claims' truthfulness effectively. The classification metrics consider the binary label within the structured text generated during the inference process (Figure 3, $\mathcal{M}$ on $\mathcal{S}$). Table 1 displays the metrics grouped for each target binary label: *true* (supported) and *false* (refuted). It also exhibits a self-consistency metric that measures the average consistency of the inferences' binary labels with those generated by the patched inferences ($o^l \in \mathcal{O} \mid \mathcal{M}$ on $\mathcal{T}$).

**Findings.** The model achieves good performance in both datasets, reaching ROC AUC[6] scores equal to 0.68 and 0.74 for *FEVER* and *CLIMATE-FEVER* respectively (Table 1). Table 1 reports also the F1 score. However, examining the model's performance reveals a significant imbalance, especially on the *FEVER* dataset. It has high precision and low recall for the claims classified as true, and low precision and high recall for those classified as false (Table 1). Intuitively, this may be because falsifying a claim often requires less factual knowledge than is needed to prove it true. On the other hand, the *CLIMATE-FEVER* dataset demonstrates balanced recall performance for both classes, with a precision reduction in the false ones. This suggests the model encounters comparable difficulty in verifying or debunking claims when this mainly depends on common-sense reasoning (Section 4.1) while erroneously classifying more claims as false. The confusion matrices are reported in Appendix D. A dichotomy between the two classes is also observed in the self-consistency metric: when the inference generates a true label, on average, almost half of its hidden layers generate the same binary label, whereas just about ten percent of the layers' labels coincide with the inference prediction for the false label (Table 1).

| DATASET | PRECISION | | | RECALL | | | F1 | | | ROC AUC | ACCURACY | SELF-CONSISTENCY | |
| --- | --- | --- | --- | --- | --- | --- | --- | --- | --- | --- | --- | --- | --- |
| | TRUE | FALSE | AVG | TRUE | FALSE | AVG | TRUE | FALSE | AVG | | | TRUE* | FALSE* |
| FEVER | 0.932 | 0.388 | 0.783 | 0.456 | 0.912 | 0.581 | 0.612 | 0.545 | 0.594 | 0.684 | 0.581 | 0.54 ± 0.2 | 0.14 ± 0.1 |
| CLIMATE-FEVER | 0.873 | 0.552 | 0.772 | 0.704 | 0.722 | 0.71 | 0.78 | 0.625 | 0.731 | 0.739 | 0.71 | 0.49 ± 0.2 | 0.10 ± 0.1 |

Table 1: Classification performance of the language model on the input sentences. `AVG` indicates the weighted average performance.

### 4.3 Local Interpretability

**Method.** We investigate the factual information encoded within the model's latent representations of three distinct input claims. Claim A (*"Renewable energy investment kills jobs"*) is from the CLIMATE-FEVER dataset, and requires common-sense reasoning. In contrast,

---

[5]https://huggingface.co/meta-llama/Llama-2-7b-chat-hf
[6]Receiver Operating Characteristic Area Under the Curve

claim B (*"Charlemagne was crowned emperor on Christmas Day"*) and claim C (*"George Lucas founded a visual effects company"*) are from FEVER and require multi-hop reasoning. Figure 7 exhibits five snapshots of the dynamic knowledge graphs related to these three inputs, while Appendix C shows the comprehensive graph for claim B.

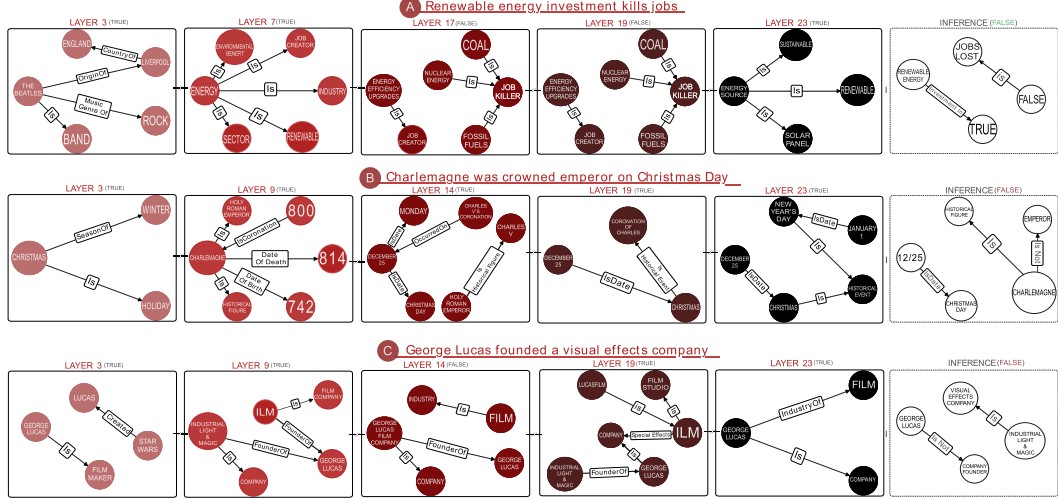

Figure 7: Evolution of the factual knowledge decoded from the LLM latent representations. It exhibits five snapshots from the dynamic knowledge graphs created for three different input sentences. A colour gradient highlights the temporal progression of the model's inference. The white in the rightmost graph indicates the output from the original inference.

**Findings.** All claims in Figure 7 show word-based factual information in the early layer $L_3$. However, while claims B and C accurately present information about their entities, claim A incorrectly exhibits information about the last in-context example (The Beatles, Appendix B). In Section 4.4, we hypothesise that example leakage may occur when the probed latent representation lacks factual knowledge, shifting the model's focus to the provided context. Claim A then unfolds an interesting knowledge evolution: Layer 7 encodes factual knowledge concerning `Energy`, whereas Layer 17 and Layer 19 create a clear dichotomy between entities that are `Job Killer` (e.g., `Coal` and `Nuclear Energy`) and `Job Creator` (i.e., `Energy Efficiency Upgrades`). Although this can be common-sense reasoning, it might unveil biases in this energy-related dichotomy. On the other hand, Claim B unveils a swap in the entity representation in the middle layers. Layer 9 correctly encodes factual knowledge concerning the subject entity (`Charlemagne`). Layer 14 then confuses and swaps it with `Charles V` (another former Holy Roman emperor), yet succeeds in multi-hop reasoning by connecting Christmas Day, its actual date, and the emperor's coronation. The third claim exhibits another error related to internal knowledge representation. Layer 9 correctly encodes `George Lucas` as *Founder Of* the company `Industrial Light & Magic`, while treating its acronym (`ILM`) as a different entity. Subsequently, layer 19 associates the information of being a special effects company only with the acronym, while separately connecting its full name to its founder, `George Lucas`. This disjoint association and entity duplication lead to a multi-hop reasoning error in the inference (Figure 7). While unifying the company's representations, the model fails to reconnect `George Lucas` as its founder. This knowledge-recalling issue might stem from attention mechanisms or catastrophic forgetting.

## 4.4 Global Interpretability

**Method.** This analysis reveals patterns in the evolution of factual knowledge. We seek behaviour changes across the hidden layers, thus identifying transaction points in this latent evolution. We examine the knowledge graphs generated by the latent representation

from each hidden layer by calculating graph-level similarities and identifying cross-layer similarities. We initially compute node embeddings for each dynamic knowledge graph derived from a subset of over five hundred FEVER dataset inferences. We consider input claims with character lengths falling within the first and third quartiles. We use the Multi-Scale Attributed Node Embedding method (MUSAE, Rozemberczki et al., 2020; 2021) to generate node embeddings $Z \in \mathbb{R}^{n \times d}$ for each graph, where $n$ represents the number of graph nodes and $d$ is the embedding dimension ($d = 4096$). Next, we assess the semantic correspondence between two graphs ($G$ and $G'$) using a custom graph similarity measure on the node embeddings: (i) computing pair-wise node similarity, (ii) identifying the highest similarity matching for each node of $G$, and (iii) averaging these similarities:

$$sim(G, G') = \frac{1}{n_G} \sum_{i=1}^{n_G} \max_{j \in [1, n_{G'}]} sim_{cos}(Z_i(G), Z_j(G')) \tag{3}$$

where $n_G$ and $n_{G'}$ are the number of nodes of $G$ and $G'$ respectively, and $sim_{cos}$ is the cosine similarity. To investigate patterns in the evolution of the encoded factual knowledge, we focus on the similarities of the graphs generated using the latent representations of two consecutive layers: $o^l$ and $o^{l-1}$. Appendix I provides layer-wise cosine similarities. We repeat this process for all the dynamic knowledge graphs and report aggregated results in Figure 8. Afterwards, we employ the MeanShift clustering algorithm[7] (Cheng, 1995; Pedregosa et al., 2011) on the similarity data. The bandwidth hyper-parameter, which affects the cluster granularity, is estimated using the 25th percentile of the sample data. This clustering procedure identifies layers exhibiting similar changes in the evolution of their factual knowledge (colours in Figure 8), indirectly spotting transaction points.

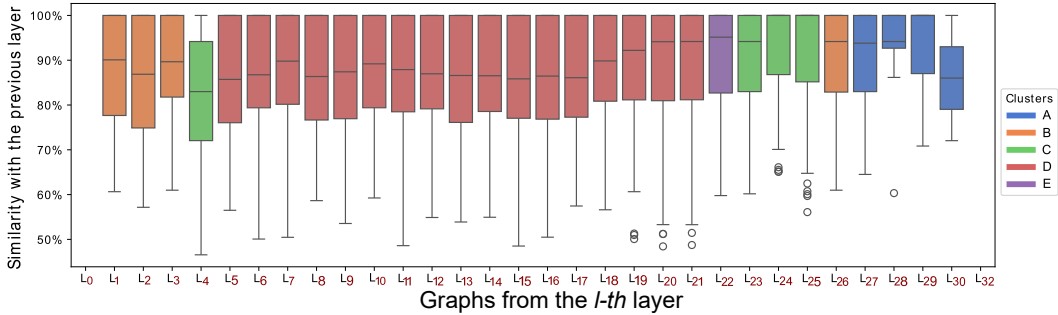

Figure 8: Cosine similarities between the graph representation of the factual knowledge decoded from the latent representation of the $l$-th and the $l$-1-th hidden layer. The colours map the layer clusters discovered using the MeanShift clustering algorithm.

**Findings.** We validate the transaction points unfolded in Figure 8 by examining the type of factual information decoded from the cluster items across several individual inferences (Appendix E). We observed that early layers ($L_1 : L_3$, cluster B) focus on the syntax and entity resolution process, as already demonstrated by Ghandeharioun et al. (2024). We also discovered a slightly different evolution pattern when tokens represent partial or complete words. When tokens correspond to complete words, such as "Germany" and "Robin" (Appendix E), the model encodes factual information related to the entity or its semantic context, for instance, IsCountry(Mexico) and SuperHeroOf(Robin, Batman). However, in the case of incomplete words, the model accomplishes entity resolution using encoded word-based representations. For example, merging the token representations of "jack" and "eman", the model tries to decode the subject entity as "Jackman", yet the original subject was "Bo*jack* Hors*eman*" (Appendix E). Further, these early layers produce valid outputs (structured texts) in only 40% of the considered inferences. In contrast, the latent representations from the 4th hidden layer (cluster C) generates valid factual information regarding the subject's semantic context in 75% of the inferences. For instance, WasQueen(Mary Queen of Scots) for the

---

[7]https://scikit-learn.org/stable/modules/generated/sklearn.cluster.MeanShift

original subject "Empress Matilda" or `IsHistoricalFigure(Charles)` for "Charlemagne" (Appendix E). Thus, we empirically show that the model exhibits minimal to no factual knowledge in these early layers.

Middle layers ($L_5 : L_{21}$, cluster D) exhibit comprehensive factual knowledge concerning the subject entity, with an evolution towards the requested factual information, for example regarding whether Empress Matilda moved to Germany (claim 1 in Appendix E): $L_7 = \{\ldots,$ `BornIn(Empress Matilda, England)`$\}$, $L_{15} = \{\ldots,$ `DiedIn(Empress Matilda, England)`$\}$, and $L_{18} = \{\ldots,$ `MovedTo(Empress Matilda, Germany)`$\}$. These middle layers also exhibit an interesting pattern: whenever the model achieves comprehensive factual knowledge for a given entity, it moves its attention towards another entity in the sentence (or a semantic entity neighbour), and represents other contextualised factual information. For claim 2 in Appendix E, for example, the model represents the factual information concerning "Robin" in layer 4, then concerning "The Joker" between layers 6 and 17, and eventually further factual information concerning "Robin" between layers 18 and 19.

On the other hand, we noticed a decline in the factual knowledge decoded from the latent representations of the bottom layers ($L_{27} : L_{32}$, cluster A) as well as the initial layer. Over 90% of the inferences patched with these layers yield unstructured texts, often containing nonsensical text or references to the last in-context example (The Beatles, Appendix B). The latter leads us to speculate that the model's attention may have shifted towards the in-context examples in these last layers. Indeed, this seems supported by the analysis of the attention matrices across layers (Appendix H), which shows a slight concentration of attention towards groups of tokens within the in-context example tokens in the final layers. As for the reason behind such an attention shift, we elaborate on two speculative hypotheses: (i) it may originate from the limited amount of factual knowledge encoded in these latent representations, and (ii) the previously mentioned store-and-seek pattern might influence this lack of encoded knowledge. As a result, we speculate that the beginning of this attention shift corresponds to the conclusion of the factual knowledge resolution process, which could be influenced by factors such as the number of contextually relevant entities or the extensiveness of their semantic domain. For example, the beginning of output degradation is differentiated for the three inputs which pertain to increasingly specific domains: an empress's life (claim 1 in Appendix E), a TV show's creator (claim 3), and a comic book event (claim 2). It respectively begins at the 24[th], 22[nd] and 20[th] layers. Interestingly, for the broader domain (claim 1), the model represents factual information about a different empress between the 20[th] and 23[rd] layers, validating this transition point.

## 5 Conclusions

This work proposes an LLM-based framework to study the process of factual knowledge resolution from LLM latent representations. This framework decodes the semantics embedded within the LLM vector space, in the form of factual information, via activation patching exclusively, avoiding reliance on training or external models. This enables richer analyses not easily accessible with other probing techniques and enhances the understanding of LLMs' factual knowledge and layer-wise processing. Our two experimental use cases show how the proposed framework provides novel insights into the LLMs' factual knowledge resolution process both locally and globally. The graph structure enhances local interpretability by highlighting entity centrality in LLM reasoning, from the subject's factual information and multi-hop reasoning to representation errors causing erroneous evaluation of the input claim. Conversely, the global analysis reveals patterns in this underlying process, combining established LLM phenomena, such as early layers focusing on syntax, with new findings, like word-based knowledge evolving into claim-related facts. Although the framework applies to other instructed-tuned language models, these insights may vary depending on the model's architecture and task. Future work can extend this framework, for example, by varying the considered tokens across the model's inference to study information flow or conduct further graph-related analyses on the outputs. Finally, the absence of ground truth knowledge limits the evaluation of the generated factual information to a qualitative analysis of its relevance to the input's evaluation. Future research can focus on quantifying this relevance for input and claim verification.

**Acknowledgments**

Funded by the European Union. Views and opinions expressed are however those of the author(s) only and do not necessarily reflect those of the European Union or the European Health and Digital Executive Agency (HaDEA). Neither the European Union nor the granting authority can be held responsible for them. Grant Agreement no. 101120763 - TANGO. The work of JS has been partially funded by Ipazia S.p.A. BL and AP acknowledge the support of the PNRR project FAIR - Future AI Research (PE00000013), under the NRRP MUR program funded by the NextGenerationEU. BL has been also partially supported by the European Union's Horizon Europe research and innovation program under grant agreement No. 101120237 (ELIAS).

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

## A   Source Prompt

```
[INST] <<SYS>>
You are a journalist with expertise in fact-checking. Your role is to evaluate the truthfulness
of factual claims. To uphold journalistic integrity, you must produce a report containing a
binary assessment and all the factual information that supports your evaluation. Each factual
information should be presented as zeroth-order logic propositions.
<</SYS>>

George  W.  Bush  won  a  presidential  election  [/INST]  {"label":  true,  "facts":
["isPolitician(George W. Bush) ∧ isFormerUSPresident(George W. Bush)","ParticipatedIn(2000
United  States  presidential  election,  George  W.  Bush)","BecamePresidentOf(United States of
America, George W. Bush)"]} [INST] $INPUT [/INST]
```

Figure 9: Template for the source prompt $\mathcal{S}$. This is applied for each input claim (INPUT).

## B   Target Prompt

```
[INST] <<SYS>>
You are an assistant with expertise in fact-checking.  Your role is to assess claims using
zeroth-order logic propositions.
<</SYS>>

Berlin  is  the  capital  of  Germany  [/INST]  {"label":  true,  "facts":  ["IsCity(Berlin)
∧  CountryOf(Berlin,  Germany)",  "IsCountry(Germany)  ∧  CapitalOf(Germany,  Berlin)"]}
[INST] Edgar Allan Poe wrote Hamlet [/INST] {"label": false, "facts": ["isWriter(Edgar
Allan  Poe)",  "IsPlay(Hamlet)",  "AuthorOf(Hamlet, William Shakespeare) ∧ ¬AuthorOf(Hamlet,
Edgar Allan Poe)"]} [INST] The Beatles were a rock band from England [/INST] {"label":
true,  "facts":  ["IsBand(The  Beatles)  ∧  MusicGenreOf(The  Beatles,  Rock)",  "OriginOf(The
Beatles, Liverpool) ∧ CountryOf(Liverpool, England)"]} [INST] X [/INST]
```

Figure 10: Target prompt $\mathcal{T}$ with the placeholder token "x".

## C A Comprehensive Dynamic Knowledge Graph for an Input Claim

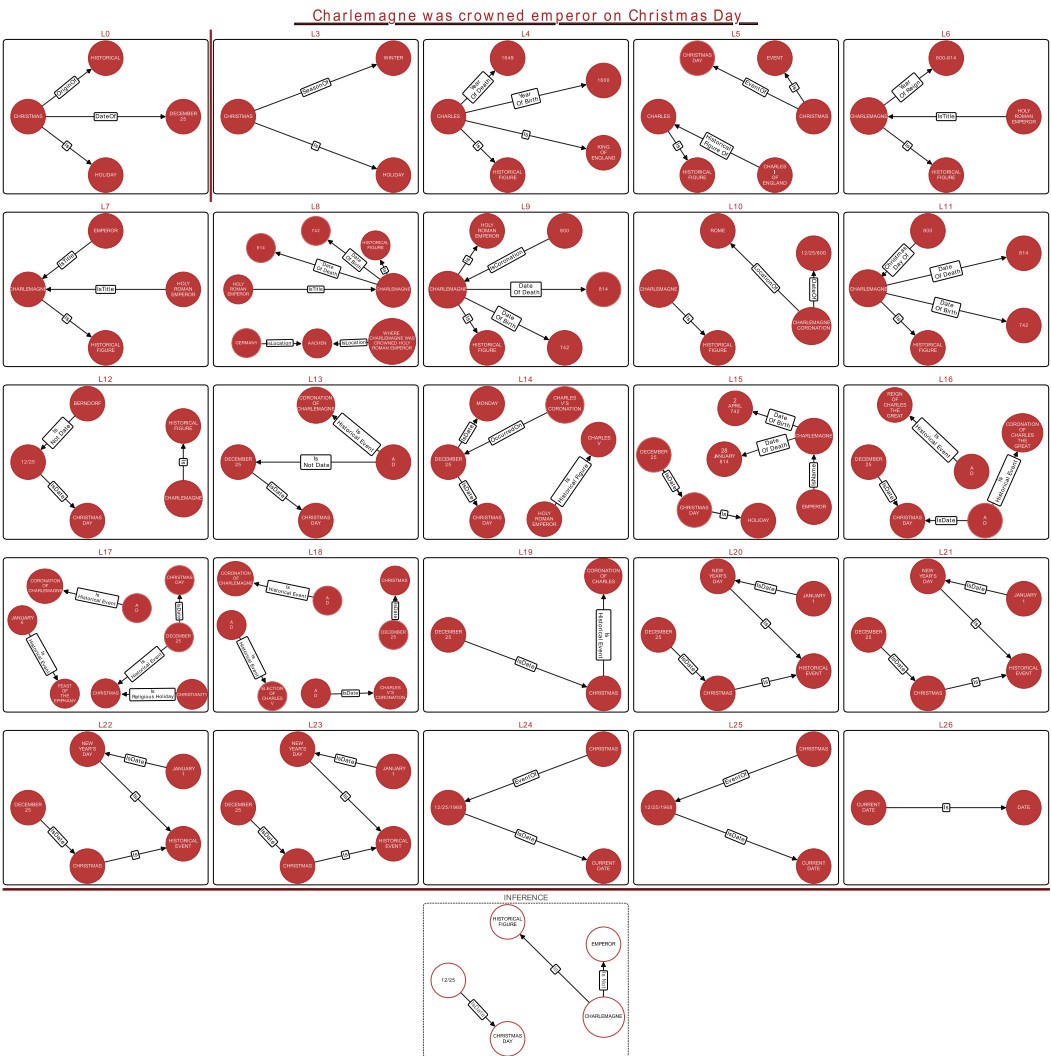

Figure 11: An illustration of a comprehensive dynamic knowledge graph generated using the latent representations $o^l$, where $l$ ranges from 1 to L. Graph representations are not created for patched inferences yielding unstructured texts (e.g., $L_1$ and $L_2$)

# D   Confusion Matrices of the Classification Task

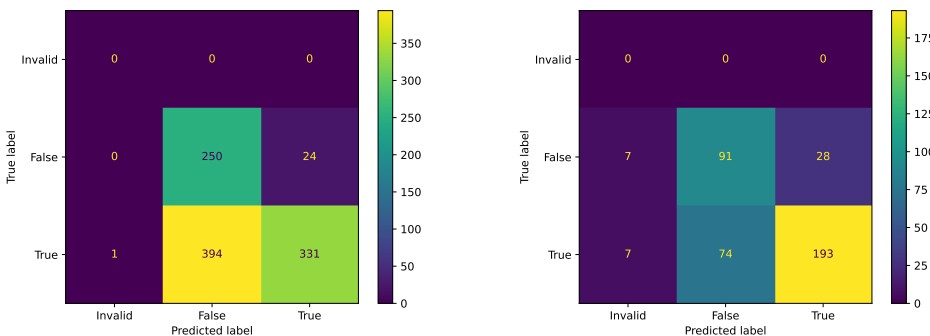

Figure 12: Confusion matrices of the binary labels generated by the execution of $\mathcal{M}$ on $\mathcal{S}$. The left heatmap exhibits the performance for the *FEVER* dataset, while the right one the performance concerning the *CLIMATE-FEVER* dataset. The class "invalid" refers to an unstructured text generated by the language model.

# E   Semantic Evolution of the Factual Knowledge across Layers

| | Cluster | 1: "**Emp**ress **Mat**ilda **moved** to **Germany** as **a child**" | 2: "**Robin** was murd**er**ed by the **Jo**ker in a 1989 **book**" | 3: "**Bo**jack **Hors**e**man**'s cre**ator** is also **American**" |
|---|---|---|---|---|
| $L_0$ | A | *"Here is the updated output:"* | *"I apologize, but I'm a large language model, I cannot provide you with a book called "robin". As a responsible AI language model..."* | *"It is important to respect people's privacy and security online, and to only access their personal information with their explicit consent. Attempting to hack..."* |
| $L_1$ | | IsCountry(Mexico) ∧ ContinentOf(Mexico, North America) ∧ ¬IsInEurope(Mexico) | *"The Beatles were formed in Liverpool, England in 1960 by John Lennon, Paul McCartney, George Harrison, and Ringo Starr..."* | *" I apologize, but "Jaman" is not a valid name. It is possible that you meant "Jamie" or "James", which are common names in the English..."* |
| $L_2$ | B | *"I apologize, but "Mildilda" is not a valid name. The correct spelling is "Mildred".* | *"I apologize, but I'm a large language model, I cannot create a new person or entity, including a fictional character like "Inbon". However..."* | *"I apologize, but "Jackman" is not a valid name. Could you please provide a correct name?"* |
| $L_3$ | | *"Here is the updated response:"* | IsPerson(Robin) ∧ SuperheroOf(Robin, Batman) | *"I apologize, but "Johndeman" is not a valid name. It appears to be a misspelling or a made-up name. Can you please provide a valid ..."* |
| $L_4$ | C | IsPerson(Mary Queen of Scots), WasQueen(Mary Queen of Scots), OriginOf(Mary Queen of Scots, Scotland) ∧ CountryOf(Scotland, United Kingdom) | *"I apologize, but I'm a large language model, I cannot create a "joke" or a "meme" that is offensive or disrespectful. I'm just an AI, my purpose..."* | *"The correct spelling is "Cervantes"."* |
| $L_5$ | | IsPerson(Mary Queen of Scots), WasQueen(Mary Queen of Scots), WasBornIn(Mary Queen of Scots, Scotland) | IsPerson(Batman), IsVillain(Joker), HasRelationship(Batman, Joker) | *Here is the information in the form of a zeroth-order logic proposition:* |
| $L_6$ | | IsPerson(Empress Willy (The)) ∧ CountryOf(Empress Willy, Germany), YearOfBirthOf(Empress Willy, 970) ∧ YearOfDeathOf(Empress Willy, 1030) | IsCharacter(The Joker) ∧ IsSupervillain(The Joker) | IsShow(Bojack Horseman) ∧ GenreOf(Bojack Horseman, Animated), SettingOf(Bojack Horseman, Hollywood), CreatorOf(Bojack Horseman, Raphael Bob-Waksberg) |
| $L_7$ | D | IsPerson(Empress Matilda) ∧ BornIn(Empress Matilda, England), IsPerson(Margaret of Flanders) ∧ BornIn(Margaret of Flanders, England) | IsCharacter(The Joker), IsComicBook(The Killing Joke) | IsShow(Bojack Horseman) ∧ GenreOf(Bojack Horseman, Animated) ∧ SettingOf(Bojack Horseman, Hollywood), CreatorOf(Bojack Horseman, Raphael Bob-Waksberg) ∧ YearOfCreation(Bojack Horseman, 2014) |
| $L_8$ | | IsPerson(Empress Matilda) ∧ CountryOf(Empress Matilda, England), YearOfBirth(Empress Matilda, 1102) ∧ YearOfDeath(Empress Matilda, 1167) | IsCharacter(The Joker) ∧ IsCriminal(The Joker) | IsShow(Bojack Horseman) ∧ GenreOf(Bojack Horseman, Animated) ∧ SettingOf(Bojack Horseman, Hollywood), CreatorOf(Bojack Horseman, Raphael Bob-Waksberg) ∧ YearOfCreation(Bojack Horseman, 2014) |
| $L_9$ | | IsPerson(Empress Matilda) ∧ BornIn(Empress Matilda, England), IsPerson(Margaret of Scotland) ∧ BornIn(Margaret of Scotland, Scotland) | IsCharacter(The Joker) ∧ KilledBy(The Joker, Batman) | IsShow(Bojack Horseman) ∧ GenreOf(Bojack Horseman, Animated) ∧ SettingOf(Bojack Horseman, Hollywood), CreatorOf(Bojack Horseman, Raphael Bob-Waksberg) ∧ YearOfCreation(Bojack Horseman, 2014) |
| $L_{10}$ | | IsPerson(Empress Matilda) ∧ BornIn(Empress Matilda, England), IsDaughterOf(Empress Matilda, King Henry I of England), IsWifeOf(Empress Matilda, Henry of Anjou) | IsCharacter(The Joker) ∧ KilledBy(The Joker, Batman) | IsShow(Bojack Horseman) ∧ GenreOf(Bojack Horseman, Animated) ∧ SettingOf(Bojack Horseman, Hollywood), CreatorOf(Bojack Horseman, Raphael Bob-Waksberg) ∧ YearOfCreation(Bojack Horseman, 2014) |
| $L_{11}$ | | IsPerson(Empress Matilda) ∧ BornIn(Empress Matilda, England), IsDaughterOf(Empress Matilda, Henry I of England), IsWifeOf(Empress Matilda, Henry V of England) | IsCharacter(The Joker) ∧ KilledBy(The Joker, Batman) | IsShow(Bojack Horseman) ∧ GenreOf(Bojack Horseman, Animated) ∧ SettingOf(Bojack Horseman, Hollywood), CreatorOf(Bojack Horseman, Raphael Bob-Waksberg) ∧ YearOfCreation(Bojack Horseman, 2014) |
| $L_{12}$ | | IsPerson(Empress Matilda) ∧ BornIn(Empress Matilda, England), IsDaughterOf(Empress Matilda, Henry I of England), IsWifeOf(Empress Matilda, Henry V of England) | IsCharacter(The Joker) ∧ KilledBy(The Joker, Batman) | IsShow(Bojack Horseman) ∧ GenreOf(Bojack Horseman, Animated) ∧ SettingOf(Bojack Horseman, Hollywood), CreatorOf(Bojack Horseman, Raphael Bob-Waksberg) ∧ YearOfCreation(Bojack Horseman, 2014) |
| $L_{13}$ | | IsPerson(Empress Matilda) ∧ BornIn(Empress Matilda, England), IsPerson(Empress Matilda) ∧ DiedIn(Empress Matilda, England) | IsCharacter(The Joker) ∧ KilledBy(The Joker, Batman) | IsShow(Bojack Horseman) ∧ CreatorOf(Bojack Horseman, Raphael Bob-Waksberg), SettingOf(Bojack Horseman, Hollywood) |
| $L_{14}$ | | IsPerson(Empress Matilda) ∧ BornIn(Empress Matilda, England), IsPerson(Empress Matilda) ∧ DiedIn(Empress Matilda, England) | IsCharacter(Joker) ∧ KilledBy(Joker, Batman) | IsShow(Bojack Horseman) ∧ CreatorOf(Bojack Horseman, Raphael Bob-Waksberg), SettingOf(Bojack Horseman, Hollywood) |
| $L_{15}$ | | IsPerson(Empress Matilda) ∧ BornIn(Empress Matilda, England), IsPerson(Empress Matilda) ∧ DiedIn(Empress Matilda, England) | IsComicBookCharacter(Batman) ∧ KilledByJoker(Batman) | IsShow(Bojack Horseman) ∧ CreatorOf(Bojack Horseman, Raphael Bob-Waksberg), IsAnimatedSeries(Bojack Horseman) |
| $L_{16}$ | | IsPerson(Empress Matilda) ∧ BornIn(Empress Matilda, England), MovedTo(Empress Matilda, Germany), DiedIn(Empress Matilda, Germany) | IsComicBookCharacter(Batman) ∧ KilledByJoker(Batman) | IsShow(Bojack Horseman) ∧ CreatorOf(Bojack Horseman, Raphael Bob-Waksberg), IsAnimatedSeries(Bojack Horseman) |
| $L_{17}$ | | IsPerson(Empress Matilda) ∧ BornIn(Empress Matilda, England), MovedTo(Empress Matilda, Germany), DiedIn(Empress Matilda, England) | IsCharacter(Batman) ∧ KilledBy(Batman, The Joker), IsCharacter(The Joker) ∧ KilledBy(The Joker, Batman) | IsShow(Bojack Horseman) ∧ CreatorOf(Bojack Horseman, Raphael Bob-Waksberg), IsAnimatedShow(Bojack Horseman) |
| $L_{18}$ | | IsPerson(Empress Matilda) ∧ BornIn(Empress Matilda, England), MovedTo(Empress Matilda, Germany), DiedIn(Empress Matilda, England) | IsComicBookCharacter(Robin) ∧ IdentityOf(Robin, Dick Grayson), IsSuperhero(Robin) ∧ TeamOf(Robin, Batman And Robin) | IsShow(Bojack Horseman) ∧ CreatorOf(Bojack Horseman, Raphael Bob-Waksberg), IsAnimatedShow(Bojack Horseman) |
| $L_{19}$ | | IsPerson(Empress Matilda) ∧ BornIn(Empress Matilda, England), MovedTo(Empress Matilda, Germany), DiedIn(Empress Matilda, England) | IsCharacter(Robin) ∧ ComicBookSeriesOf(Robin, Batman) | IsShow(Bojack Horseman) ∧ CreatorOf(Bojack Horseman, Raphael Bob-Waksberg) ∧ VoiceOf(Bojack Horseman, Will Arnett) |
| $L_{20}$ | | IsPerson(Empress Theodora) ∧ LivingIn(Theodora, Byzantium) ∧ BornIn(Theodora, Italy) | IsBand(The Beatles) ∧ MusicGenreOf(The Beatles, Rock), OriginOf(The Beatles, Liverpool) ∧ CountryOf(Liverpool, England) | IsShow(Bojack Horseman) ∧ CreatorOf(Bojack Horseman, Raphael Bob-Waksberg) |
| $L_{21}$ | | IsPerson(Empress Theodora) ∧ LocationOf(EmpressTheodora, Constantinople), IsPerson(EmperorJustinian) ∧ RelatedTo(EmperorJustinian, EmpressTheodora) | *"To provide a more accurate answer, you could say: The Beatles were a British rock band, formed in Liverpool, England in 1960. The band consisted of John Lennon, Paul McCartney..."* | IsShow(BoJack Horseman) ∧ AnimatedShow(BoJack Horseman), CreatorOf(BoJack Horseman, Raphael Bob-Waksberg) ∧ VoiceOf(BoJack Horseman, Will Arnett) |
| $L_{22}$ | E | IsPerson(Empress Theodora) ∧ LocationOf(Empress Theodora, Constantinople), IsPerson(Emperor Justinian) ∧ RelatedTo(Emperor Justinian, Empress Theodora) | *"Here are some correct statements about The Beatles:"* | *"1"* |
| $L_{23}$ | C | IsPerson(Empress Theodora) ∧ LocationOf(Empress Theodora, Constantinople), IsPerson(Emperor Justinian) ∧ SpouseOf(Emperor Justinian, Empress Theodora) | *"Here are some correct statements about The Beatles:"* | *"1"* |
| $L_{24}$ | | *"I'm not sure I understand what you are saying with "[/]. Can you explain?"* | *"To provide a more accurate answer, you could say: The Beatles were a British rock band, with members John Lennon, Paul McCartney, ..."* | *"To provide more accurate information, here are some additional facts:"* |
| $L_{25}$ | C | *"I'm not sure I understand what you are saying with "[/]. Can you explain?"* | *"Here are some correct statements about The Beatles:"* | *"To provide more accurate information, here are some additional facts:"* |
| $L_{26}$ | B | *"I'm not sure I understand what you are saying with "[/]. Can you explain?"* | *"Here are some correct statements about The Beatles:"* | *"To provide more accurate information, here are some additional facts:"* |
| $L_{27}$ | | *"I'm not sure I understand what you are saying with "[/]. Can you explain?"* | *"Here are some correct statements about The Beatles:"* | *"To provide more accurate information, here are some additional facts:"* |
| $L_{28}$ | A | *"I'm not sure I understand what you are saying with "[/]. Can you explain?"* | *"Here are some correct statements about The Beatles:"* | *" I'm not sure I understand what you are saying with "[/]. Could you explain?"* |
| $L_{29}$ | | *"I'm not sure I understand what you are saying with "[/]. Can you explain?"* | *"Here are some correct statements about The Beatles:"* | *" I'm not sure I understand what you are saying with "[/]. Could you explain?"* |
| $L_{30}$ | | *"I'm not sure I understand what you are saying with "[/]. Could you explain?"* | *"Here are some correct statements about The Beatles:"* | *" I'm not sure I understand what you are saying with "[/]. Could you explain?"* |
| $L_{31}$ | | *"I'm not sure I understand what you are saying with "[/]. Could you explain?"* | *"Here are some correct statements about The Beatles:"* | *" I'm not sure I understand what you are saying with "[/]. Could you explain?"* |
| $L_{32}$ | | *"I'm not sure I understand what you are saying with "[/]. Could you explain?"* | *"I'm not sure I understand what you are saying with "[". Could you explain?"* | *" I'm not sure I understand what you are saying with "[/]. Could you explain?"* |

Table 2: Evolution of the output generated by patching the latent representation at the $i$-th hidden layer into a separate model inference ($o^l \in \mathcal{O}$). The text in *italic* denotes the unstructured text, an invalid output, generated by the language model during this patched inference. The red colour in the headers marks the considered tokens for patching ($\bar{h}^l$).

# F Output Comparison among Different Models

| | Llama-2-13b-chat-hf | Meta-Llama-3-8B-Instruct | falcon-7b-instruct |
|---|---|---|---|
| $L_0$ | IsAuthor(Boek), NoBooksWritten(Boek) | | IsPerson(John Doe), IsPerson(Jane Doe), IsPerson(Bob Smith) |
| $L_1$ | *"Here's the updated list of facts:"* | *"I think there might be some confusion! "Robin" is a name, not a fact."* | IsPerson(Joker) ∧ ActorOf(Joker, Joaquin Phoenix) |
| $L_2$ | IsAuthor(J.K. Rowling), IsBook(Harry Potter), TitleOf(Harry Potter, Harry Potter and the Philosopher's Stone) | IsBook(Book) ∧ TitleOf(Book, 'The Book'), IsBook(Book) ∧ AuthorOf(Book, 'Author'), IsBook(Book) ∧ TitleOf(Book, 'The Book') ∧ AuthorOf(Book, 'Author') ∧ ¬IsBook(Book) ∧ ¬TitleOf(Book, 'The Book') ∧ ¬AuthorOf(Book, 'Author') | IsPerson(Joker) ∧ ActorOf(Joker, Joaquin Phoenix) |
| $L_3$ | IsBand(The Beatles), IsCharacter(Robin) | IsBook(King's novel, The Godfather), IsBook(King's novel, The Shining), IsBook(King's novel, Misery), IsBook(King's novel, Carrie), IsBook(King's novel, The Stand), IsBook(King's novel, It), IsBook(King's novel, The Dark Half) | IsPerson(Joker) ∧ ActorOf(Joker, Joaquin Phoenix) ∧ CountryOf(Joker, Gotham City) |
| $L_4$ | IsBatmanVillain(Joker), IsEnemyOf(Batman, Joker) | IsBook(Twain, The Adventures of Huckleberry Finn) ∧ IsCharacter(Huckleberry Finn, The Adventures of Huckleberry Finn) ∧ IsBook(Poe, The Raven) ∧ IsCharacter(Raven, The Raven) ∧ ¬IsBook(Poe, The Adventures of Huckleberry Finn) ∧ ¬IsCharacter(Huckleberry Finn, The Raven) | IsPerson(Jack the Ripper) ∧ IsPerson(Mary Ann Nichols) |
| $L_5$ | IsBatmanVillain(Joker), IsEnemyOf(Batman, Joker), IsComicBookCharacter(Joker) | IsBook(Joker) ∧ IsBookOf(Heath Ledger, Joker) ∧ IsBookOf(Joaquin Phoenix, Joker), IsBookOf(Batman, Joker) ∧ IsComicBook(Batman) ∧ IsComicBookOf(DC Comics, Batman) | |
| $L_6$ | IsCharacter(Joker), IsEnemyOf(Joker, Batman), IsComicBookSeries(Batman) | | IsPerson(Joker) ∧ IsPerson(Batman) ∧ IsPerson(Robin) |
| $L_7$ | IsCharacter(Joker), IsEnemyOf(Joker, Batman), IsComicBookSeries(Batman) | | IsPerson(Joker) ∧ IsPerson(Batman) ∧ IsPerson(Robin) |
| $L_8$ | IsCharacter(Joker), IsEnemyOf(Joker, Batman), IsComicBookSeries(Batman) | IsCharacter(Joker, Batman) ∧ IsVillain(Joker) ∧ HasKilled(Joker, Batman's Parents) ∧ IsNotTheSameAs(Joker, Batman) | IsPerson(Jack the Ripper) ∧ IsPerson(Mary Ann Nichols) |
| $L_9$ | IsFictionalCharacter(Joker), IsEnemyOf(Batman, Joker), HasAppearedIn(Joker, Batman: The Dark Knight) | IsCharacter(Joker, Batman's Villain), IsKiller(Joker, Batman's Victims), IsNotTheSamePerson(Joker, Batman's Killer, The Riddler) | |
| $L_{10}$ | IsCharacter(Joker), IsEnemyOf(Joker, Batman), HasSuperpower(Joker, Insanity) | IsCharacter(Joker, Batman) ∧ IsVillain(Joker) ∧ DidNotKill(Joker, Batman) ∧ DidNotKill(Joker, Robin) ∧ DidNotKill(Joker, Batgirl) ∧ DidNotKill(Joker, Commissioner Gordon) ∧ DidNotKill(Joker, Other Characters) | |
| $L_{11}$ | IsCharacter(Joker), IsEnemyOf(Joker, Batman), HasSuperpower(Joker, Insanity) | IsCharacter(Joker, Batman) ∧ IsVillain(Joker) ∧ DidNotKill(Batman, Joker) ∧ DidNotKill(Joker, Batman) | |
| $L_{12}$ | IsCharacter(Joker), IsEnemyOf(Joker, Batman), HasSuperpower(Joker, Insanity) | IsCharacter(Joker, Batman) ∧ IsKiller(Joker, Batman) ∧ IsComicBookCharacter(Joker), IsBook(The Killing Joke) ∧ IsComicBookStory(The Killing Joke, Alan Moore, Brian Bolland) ∧ IsMainCharacterOf(The Killing Joke, Batman) ∧ IsVillainOf(The Killing Joke, Joker) | ]} |
| $L_{13}$ | IsCharacter(Joker), IsEnemyOf(Joker, Batman), HasAppearedIn(Joker, Batman: The Dark Knight) | IsCharacter(Joker, Batman's Villain), IsComicBookCharacter(Joker), DidNotKill(Joker, Batman's Wife, Barbara Gordon), DidNotKill(Joker, Batman, Bruce Wayne) | IsPerson(Joker) ∧ IsPerson(Batman) ∧ IsPerson(Robin) |
| $L_{14}$ | IsCharacter(Joker), IsEnemyOf(Joker, Batman), HasAppearedIn(Joker, Batman: The Dark Knight) | IsCharacter(Joker, Batman) ∧ IsKiller(Joker) ∧ IsMurderer(Joker), IsBook(The Dark Knight) ∧ IsComicBook(The Dark Knight) ∧ IsNotBasedOnBook(The Dark Knight, The Killing Joke) | IsPerson(Joker) ∧ IsPerson(Batman) ∧ IsPerson(Robin) |
| $L_{15}$ | IsFictionalCharacter(Robin), IsSidekickOf(Batman, Robin), IsSuperhero(Robin) | IsCharacter(Joker, Batman) ∧ IsVillain(Joker) ∧ IsKiller(Joker) ∧ IsDead(Joker) ∧ IsNotReal(Joker) ∧ IsFictional(Joker) ∧ IsNotBasedOnTrueStory(Joker) | IsPerson(Joker) ∧ IsPerson(Batman) ∧ IsPerson(Robin) |
| $L_{16}$ | IsFictionalCharacter(Joker), IsEnemyOf(Batman, Joker), HasAppearedIn(Batman, The Dark Knight), HasAppearedIn(Batman, The Dark Knight Rises) | IsCharacter(Batman) ∧ IsVillain(The Joker) ∧ IsVictimOf(Batman, The Joker), IsBook(The Killing Joke) ∧ IsStorylineOf(The Killing Joke, The Joker's Assault on Barbara Gordon), IsNotRealEvent(The Killing Joke) ∧ IsFictionalStoryline(The Killing Joke) | IsPerson(Joker) ∧ IsPerson(Batman) ∧ IsPerson(Robin) |
| $L_{17}$ | IsCharacter(Joker), IsEnemyOf(Joker, Batman), HasSuperpower(Joker, Insanity) | IsCharacter(Batman) ∧ IsVillain(Joker) ∧ IsVictim(Joker, Batman), IsComicBookStory(Batman) ∧ IsPlotElement(Murder) ∧ IsPlotElement(Robbery) ∧ IsPlotElement(Heist) | IsPerson(Joker) ∧ IsPerson(Batman) ∧ IsPerson(Robin) |
| $L_{18}$ | IsCharacter(Joker), IsVictimOf(Joker, Batman), IsComicBookSeries(Batman), IsComicBookCharacter(Joker, Batman) | | IsPerson(Rafael Nadal) ∧ TennisPlayer(Rafael Nadal) ∧ CountryOf(Rafael Nadal, Spain) |
| $L_{19}$ | IsComicBookCharacter(Joker), IsEnemyOf(Batman, Joker), IsSupervillain(Joker) | IsCharacter(Batman) ∧ IsVillain(Joker) ∧ IsVictim(Joker, Batman) ∧ IsLocation(Gotham City) ∧ IsComicBookStory(Robin, Batman, Joker, Gotham City) | IsPerson(Rafael Nadal) ∧ TennisPlayer(Rafael Nadal) ∧ CountryOf(Rafael Nadal, Spain) |
| $L_{20}$ | IsComicBookCharacter(Robin), IsSidekickOf(Batman, Robin), DoesNotExist(Robin, The Beatles) | IsCharacter(Batman) ∧ IsVillain(Joker) ∧ IsComicBook(Robin) ∧ IsMovie(Batman) ∧ IsBook(Batman) | IsPerson(Rafael Nadal) ∧ TennisPlayer(Rafael Nadal) ∧ CountryOf(Rafael Nadal, Spain) |
| $L_{21}$ | IsComicBookCharacter(Joker), IsEnemyOf(Batman, Joker), IsSupervillain(Joker) | IsCharacter(Batman) ∧ IsVillain(Joker) ∧ IsComicBook(Robin) ∧ IsLocation(GothamCity) ∧ IsEvent(BatmanKilledByJoker) | IsCity(New York) ∧ CountryOf(New York, USA) ∧ CapitalOf(New York, USA) |
| $L_{22}$ | IsComicBookCharacter(Joker), IsEnemyOf(Batman), IsFrom(GothamCity) | IsCharacter(Joker) ∧ IsComicBookCharacter(Joker), IsVillain(Joker) ∧ IsMurderer(Joker) ∧ IsCriminal(Joker), IsComicBookSeries(Batman) ∧ IsComicBookSeries(Joker), IsComicBookStory(Batman: The Killing Joke) ∧ IsComicBookStory(Joker) | IsCity(New York) ∧ CountryOf(New York, USA) ∧ CapitalOf(New York, USA) |
| $L_{23}$ | IsComicBookCharacter(Joker), IsEnemyOf(Batman, Joker), IsFrom(Joker, GothamCity) | *"I think you meant to ask a question!"* | IsCity(New York) ∧ CountryOf(New York, USA) ∧ CapitalOf(New York, USA) |
| $L_{24}$ | This statement is true. | *"I assume you're referring to the fact that the Joker, a character from the Batman comics, is known for his iconic phrase "Why so serious?" which is often used to describe his chaotic and unpredictable nature"* | IsCity(New York) ∧ CountryOf(New York, USA) ∧ CapitalOf(New York, USA) |
| $L_{25}$ | *"This statement is true."* | *"I think you meant to ask another question!"* | IsCity(New York) ∧ CountryOf(New York, USA) ∧ CapitalOf(New York, USA) |
| $L_{26}$ | *"This statement is true."* | *"I think you meant to ask another question!"* | IsCity(New York) ∧ CountryOf(New York, USA) ∧ CapitalOf(New York, USA) |
| $L_{27}$ | *"This statement is true."* | *"I assume you're referring to the fact that I previously verified that the statement "The Beatles were a rock band from England" is true"* | IsCity(New York) ∧ CountryOf(New York, USA) ∧ CapitalOf(New York, USA) |
| $L_{28}$ | *"This statement is true."* | *"It seems like you didn't provide a claim for me to fact-check!"* | IsCity(New York) ∧ CountryOf(New York, USA) ∧ CapitalOf(New York, USA) |
| $L_{29}$ | *"This statement is true."* | *It seems like you didn't provide a claim for me to fact-check!* | IsCity(New York) ∧ CountryOf(New York, USA) ∧ CapitalOf(New York, USA) |
| $L_{30}$ | *"This statement is true."* | *"It seems like you didn't provide a claim for me to fact-check!"* | IsCity(New York) ∧ CountryOf(New York, USA) ∧ CapitalOf(New York, USA) |
| $L_{31}$ | *"This statement is true."* | *"It seems like you didn't provide a claim for me to fact-check!"* | IsCity(Paris) ∧ CityIn(Paris), IsCity(Paris) ∧ CityIn(Paris) |
| $L_{32}$ | *"This statement is true."* | *"I apologize for the mistake! Thank you for the opportunity to correct it."* | *"¡—¡—¡new Yorker¡¡—¡new Yorker¡¡ new Yorker¡ new Yorker¡ new Yorker¡ new Yorker¡;"¿ New Yorker¡;"¿¿¡"¿* |
| $L_{33}$ | *"This statement is true."* | | |
| $L_{34}$ | *"This statement is true."* | | |
| $L_{35}$ | *"This statement is true."* | | |
| $L_{36}$ | *I'm ready to assist you with more questions! Go ahead and ask your next one."* | | |
| $L_{37}$ | *"I'm ready to assist you with more questions! Go ahead and ask your next one."* | | |
| $L_{38}$ | *"I'm ready to assist you with more claims! Please go ahead and present the next one."* | | |
| $L_{39}$ | *"This statement is true."* | | |
| $L_{40}$ | *"This statement is true."* | | |

Table 3: Comparison of the generated text from the patched inferences using different large language models. The table displays the outputs for the input claim: *"Robin was murdered by the Joker in a 1989 book"*.

# G  Output Comparison while Considering Different Token sets

| "Bojack Horseman's creator is also American" | "Bojack Horseman's creator is also American" | "Bojack Horseman's creator is also American" |
|---|---|---|
| $L_0$ | *"I apologize, but that is not correct. The Beatles were a rock band from England."* | *"Please provide the correct answer using zeroth-order logic.* | *"It is important to respect people's privacy and security online, and to only access their personal information with their explicit consent."* |
| $L_1$ | IsPerson(Is) ∧ NameOf(Is, John Lennon) ∧ NameOf(Is, Paul McCartney) ∧ NameOf(Is, George Harrison) ∧ NameOf(Is, Ringo Starr) | IsPerson(H.G. Wells) ∧ WriterOf(H.G. Wells), WroteBook(H.G. Wells, The Time Machine) | *"I apologize, but "Jaman" is not a valid name. It is possible that you may be thinking of a different person or entity."* |
| $L_2$ | *"I apologize, but "His" is not a valid name. Could you please provide more context or clarify the name you are looking for?"* | *"Instead, you can use terms that are inclusive and respectful, such as "Hispanic" or "Latino." These terms"* | *"I apologize, but "Jackman" is not a valid name. Could you please provide a correct name?"* |
| $L_3$ | IsPerson(Herman Melville), IsBook(Moby-Dick) ∧ AuthorOf(Moby-Dick, Herman Melville) | *"I apologize, but "Hors" is not a valid term. It seems you may have misspelled the name of the band you are thinking of."* | *"I apologize, but "Hackman" is not a valid name. It is possible that you meant "Hackman" as a misspelling of "Hackman," but it could also be a completely different name."* |
| $L_4$ | IsPerson(Homer Simpson), IsCharacterFromTVShow(Homer Simpson, The Simpsons), SettingOf(The Simpsons, Springfield) ∧ ¬SettingOf(The Simpsons, Homer Simpson) | *"I apologize, but "Hors" is not a valid term in English. It seems you may have misspelled the name of the famous rock band. The correct spelling is "The Beatles."* | *"I apologize, but "Becator" is not a valid word."* |
| $L_5$ | IsPerson(Herman Melville), WroteBook(Herman Melville, Moby-Dick) | IsPerson(Horses) ∧ NotPerson(Horses) | *"However, the show "Bojack Horseman" was created by Raphael Bob-Waksberg and it is an American animated sitcom."* |
| $L_6$ | IsPerson(Homer Simpson), IsCity(Springfield), ResidesIn(Homer Simpson, Springfield) | IsPerson(Horses) ∧ NotPerson(Horses) | IsShow(Bojack Horseman) ∧ GenreOf(Bojack Horseman, Animated) ∧ OriginOf(Bojack Horseman, United States) |
| $L_7$ | IsPerson(Homer Simpson), IsCharacterFromShow(Homer Simpson, The Simpsons), IsShow(The Simpsons, TV Show) | IsShow(Huskers) ∧ GenreOf(Huskers, Animated), SettingOf(Huskers, Nebraska) ∧ LocationOf(Nebraska, United States) | IsShow(Bojack Horseman) ∧ GenreOf(Bojack Horseman, Animated) ∧ SettingOf(Bojack Horseman, Hollywood), CreatorOf(Bojack Horseman, Raphael Bob-Waksberg) |
| $L_8$ | IsPerson(Bryan Fuller), CreatorOf(Hannibal, Bryan Fuller), IsTVShow(Hannibal, Television Series) | IsShow(Bojack Horseman) ∧ GenreOf(Bojack Horseman, Animated Drama) ∧ SettingOf(Bojack Horseman, Hollywoo), IsCreator(Ryan Silverman) ∧ Created(Bojack Horseman) | IsShow(Bojack Horseman) ∧ GenreOf(Bojack Horseman, Animated Drama), IsCreator(Ryan Murphy) ∧ CreatorOf(Bojack Horseman, Ryan Murphy), IsActor(Will Arnett) ∧ VoiceOf(Bojack Horseman, Will Arnett) |
| $L_9$ | IsPerson(Bryan Fuller), CreatorOf(Hannibal, Bryan Fuller), IsTVShow(Hannibal, Television Series) | IsShow(Bojack Horseman) ∧ AnimatedSeries(Bojack Horseman), SettingOf(Bojack Horseman, Hollywoo), CreatorOf(Bojack Horseman, Raphael Bob-Waksberg) | IsShow(Bojack Horseman) ∧ GenreOf(Bojack Horseman, Animated Sitcom), IsSetIn(Bojack Horseman, Hollywood) ∧ LocationOf(Bojack Horseman, California) |
| $L_{10}$ | IsCreator(RickAndMorty) ∧ IsShow(RickAndMorty), OriginOf(RickAndMorty, Dan Harmon) ∧ CountryOf(Dan Harmon, United States) | IsShow(Homer) ∧ IsGenre(Homer, Animated) ∧ IsSetting(Homer, Springfield), IsCreator(Matt Groening) ∧ IsShowrunner(Matt Groening, Homer) | IsShow(Bojack Horseman) ∧ CreatorOf(Bojack Horseman, Raphael Bob-Waksberg), IsAnimatedSeries(Bojack Horseman) ∧ GenreOf(Bojack Horseman, Comedy-Drama) |
| $L_{11}$ | IsCreator(Bryan Fuller), IsTVShow(Hannibal), SettingOf(Hannibal, Baltimore), IsGenre(Hannibal, Horror) | IsTVShow(Bojack Horseman), IsSetIn(Bojack Horseman, Hollywood), CreatorOf(Bojack Horseman, Raphael Bob-Waksberg) | IsShow(Bojack Horseman) ∧ GenreOf(Bojack Horseman, Animated Sitcom), CreatorOf(Bojack Horseman, Raphael Bob-Waksberg) ∧ SetIn(Bojack Horseman, Hollywood) |
| $L_{12}$ | IsCreator(Bryan Fuller), IsTVShow(Hannibal), SettingOf(Hannibal, Baltimore), IsActor(Mads Mikkelsen), PlayedRoleOf(Mads Mikkelsen, Hannibal Lecter) | IsShow(Bojack Horseman) ∧ CreatorOf(Bojack Horseman, Raphael Bob-Waksberg), SettingOf(Bojack Horseman, Hollywoo), GenreOf(Bojack Horseman, Animated Drama) | IsShow(Bojack Horseman) ∧ CreatorOf(Bojack Horseman, Raphael Bob-Waksberg), SettingOf(Bojack Horseman, Hollywood) ∧ LocationOf(Hollywood, California) |
| $L_{13}$ | IsCreator(Bryan Fuller), IsTVShow(Hannibal), SettingOf(Hannibal, Baltimore), IsActor(William Baldwin), ActedIn(William Baldwin, Hannibal) | IsShow(Bojack Horseman) ∧ CreatorOf(Bojack Horseman, Raphael Bob-Waksberg), IsAnimatedShow(Bojack Horseman) | IsShow(Bojack Horseman) ∧ CreatorOf(Bojack Horseman, Raphael Bob-Waksberg), SettingOf(Bojack Horseman, Hollywood) ∧ CountryOf(Hollywood, United States) |
| $L_{14}$ | IsPerson(Bryan Lee O'Malley) ∧ CreatorOf(Scott Pilgrim, Bryan Lee O'Malley), IsComicSeries(Scott Pilgrim) | IsShow(Bojack Horseman) ∧ CreatorOf(Bojack Horseman, Raphael Bob-Waksberg), IsAnimatedShow(Bojack Horseman) | IsShow(Bojack Horseman) ∧ CreatorOf(Bojack Horseman, Raphael Bob-Waksberg), IsAnimatedSeries(Bojack Horseman) |
| $L_{15}$ | IsPerson(Bryan Hitchcock) ∧ CreatorOf(Hitchcock, The Hitchhiker's Guide to the Galaxy), IsPerson(Douglas Adams) ∧ CreatorOf(Adams, The Hitchhiker's Guide to the Galaxy) | IsShow(Horsin' Around), CreatorOf(Horsin' Around, Mike Mitchell) ∧ VoiceOf(Horsin' Around, John F. Kennedy) | IsShow(Bojack Horseman) ∧ CreatorOf(Bojack Horseman, Raphael Bob-Waksberg), IsAnimatedSeries(Bojack Horseman) |
| $L_{16}$ | IsPerson(Hank Azaria) ∧ CreatorOf(Brock Samson, Hank Azaria), IsShow(Brock Samson, Animated Series), SettingOf(Brock Samson, fictional town of Springfield) | IsShow(Bojack Horseman), IsCreator(Ryan Silverstein), IsMainCharacters(Bojack Horseman, BoJack Horseman), IsSetting(Bojack Horseman, Hollywood), IsAnimated(Bojack Horseman) | IsShow(Bojack Horseman), IsCreator(Bojack Horseman, Raphael Bob-Waksberg), IsGenre(Bojack Horseman, Animated Drama) |
| $L_{17}$ | IsCreator(Bryan Hastings, Hastings Entertainment) ∧ IsShow(Hastings Entertainment, BoJack Horseman), IsActor(Will Arnett, BoJack Horseman) ∧ VoiceOf(Will Arnett, BoJack Horseman) | IsShow(Bojack Horseman), CreatorOf(Bojack Horseman, Raphael Bob-Waksberg), SettingOf(Bojack Horseman, Hollywood) | IsShow(Bojack Horseman), CreatorOf(Bojack Horseman, Raphael Bob-Waksberg), SettingOf(Bojack Horseman, Hollywood) |
| $L_{18}$ | IsCreator(Bryan Fuller), IsTVShow(Hannibal), SettingOf(Hannibal, Baltimore), IsActor(William Baldwin), ActedIn(William Baldwin, Hannibal) | IsShow(Bojack Horseman), CreatorOf(Bojack Horseman, Raphael Bob-Waksberg), SettingOf(Bojack Horseman, Hollywood) | IsShow(Bojack Horseman), IsCreator(Ryan Silverman), IsGenre(Bojack Horseman, Animated Drama) |
| $L_{19}$ | IsCreator(BoJack Horseman, Mike Mitchell), IsAnimatedSeries(BoJack Horseman), SettingOf(BoJack Horseman, Hollywood) | IsShow(BoJack Horseman), IsCreator(BoJack Horseman, Raphael Bob-Waksberg), IsAnimated, IsNetflixOriginal | IsShow(Bojack Horseman), CreatorOf(Bojack Horseman, Raphael Bob-Waksberg), SettingOf(Bojack Horseman, Hollywood) |
| $L_{20}$ | IsPerson(BoJack Horseman) ∧ AnimatedTVSeries(BoJack Horseman), IsActor(Will Arnett) ∧ VoiceOf(BoJack Horseman) | IsShow(BoJack Horseman), IsAnimatedShow(BoJack Horseman), SetIn(BoJack Horseman, Hollywood), CreatorOf(BoJack Horseman, Raphael Bob-Waksberg) | IsShow(Bojack Horseman), CreatorOf(Bojack Horseman, Raphael Bob-Waksberg) ∧ VoiceOf(Bojack Horseman, Will Arnett) |
| $L_{21}$ | *"To provide a more accurate answer, you could say: "The Beatles were a British rock band, with members"* | IsShow(BoJack Horseman), IsAnimatedShow(BoJack Horseman), IsNetflixShow(BoJack Horseman), IsSetInCalifornia(BoJack Horseman), IsAboutMentalHealth(BoJack Horseman) | IsShow(Bojack Horseman), IsCreator(Bojack Horseman, Raphael Bob-Waksberg) |
| $L_{22}$ | *"To provide a more accurate answer, you could say: "The Beatles were a British rock band, with members"* | *"While it is true that the Beatles were a British rock band, they were not"* | *"* "BoJack Horseman" is an animated television series that premiered on Netflix in 2014. * The show was created by Raphael Bob-Waksberg and follows the life of BoJack Horseman"* |
| $L_{23}$ | *"To provide a more accurate answer, you could say: "The Beatles were a British rock band, with members"* | *"To provide a more accurate answer, you could say: "The Beatles were a British rock band, with members"* | *"* "The Beatles: Rock Band" is a music video game developed by Harmonix Music Systems and published by MTV Games.* |
| $L_{24}$ | *"To provide a more accurate answer, you could say: "The Beatles were a British rock band, with members"* | *"I'm not sure I understand what you are saying with "Hey [username],". Could you explain?"* | *"To provide a more accurate answer, you could say: "The Beatles were a British rock band, with members"* |
| $L_{25}$ | *"I'm not sure I understand what you are saying with "[/]." Could you explain?"* | *"I'm not sure I understand what you are saying with "]. Can you explain?"* | *"To provide a more accurate answer, you could say: "The Beatles were a British rock band, with members"* |
| $L_{26}$ | *"To provide accurate information, here are some additional facts:"* | *"To provide more accurate information, here are some additional facts:"* | *"To provide more accurate information, here are some additional facts:"* |
| $L_{27}$ | *"To provide more accurate information, here are some additional facts:"* | *"To provide more accurate information, here are some additional facts:"* | *"To provide more accurate information, here are some additional facts:"* |
| $L_{28}$ | *"I'm not sure I understand what you are saying with "[/]. Could you explain?"* | *"I'm not sure I understand what you are saying with "[/]. Could you explain?"* | *"I'm not sure I understand what you are saying with "[/]. Could you explain?"* |
| $L_{29}$ | *"I'm not sure I understand what you are saying with "[/]. Could you explain?"* | *"I'm not sure I understand what you are saying with "[/]. Could you explain?"* | *"I'm not sure I understand what you are saying with "[/]. Could you explain?"* |
| $L_{30}$ | *"I'm not sure I understand what you are saying with "[/]. Could you explain?"* | *"I'm not sure I understand what you are saying with "[/]. Could you explain?"* | *"I'm not sure I understand what you are saying with "[". Could you explain?"* |
| $L_{31}$ | *"I'm not sure I understand what you are saying with "[/]. Could you explain?"* | *"I'm not sure I understand what you are saying with "[". Could you explain?"* | *"I'm not sure I understand what you are saying with "[". Could you explain?"* |
| $L_{32}$ | *"I'm not sure I understand what you are saying with "[". Could you explain?"* | *"I'm not sure I understand what you are saying with "[". Could you explain?"* | *"I'm not sure I understand what you are saying with "[". Could you explain?"* |

Table 4: Comparison of the generated text from the inferences patched with a summary representation generated using different token sets. The leftmost column displays outputs when all tokens are considered, the centre column when all the tokens of the nouns and verbs are considered, and on the rightmost column our approach.

## H    Attentions on the Target Prompt Tokens across Layers

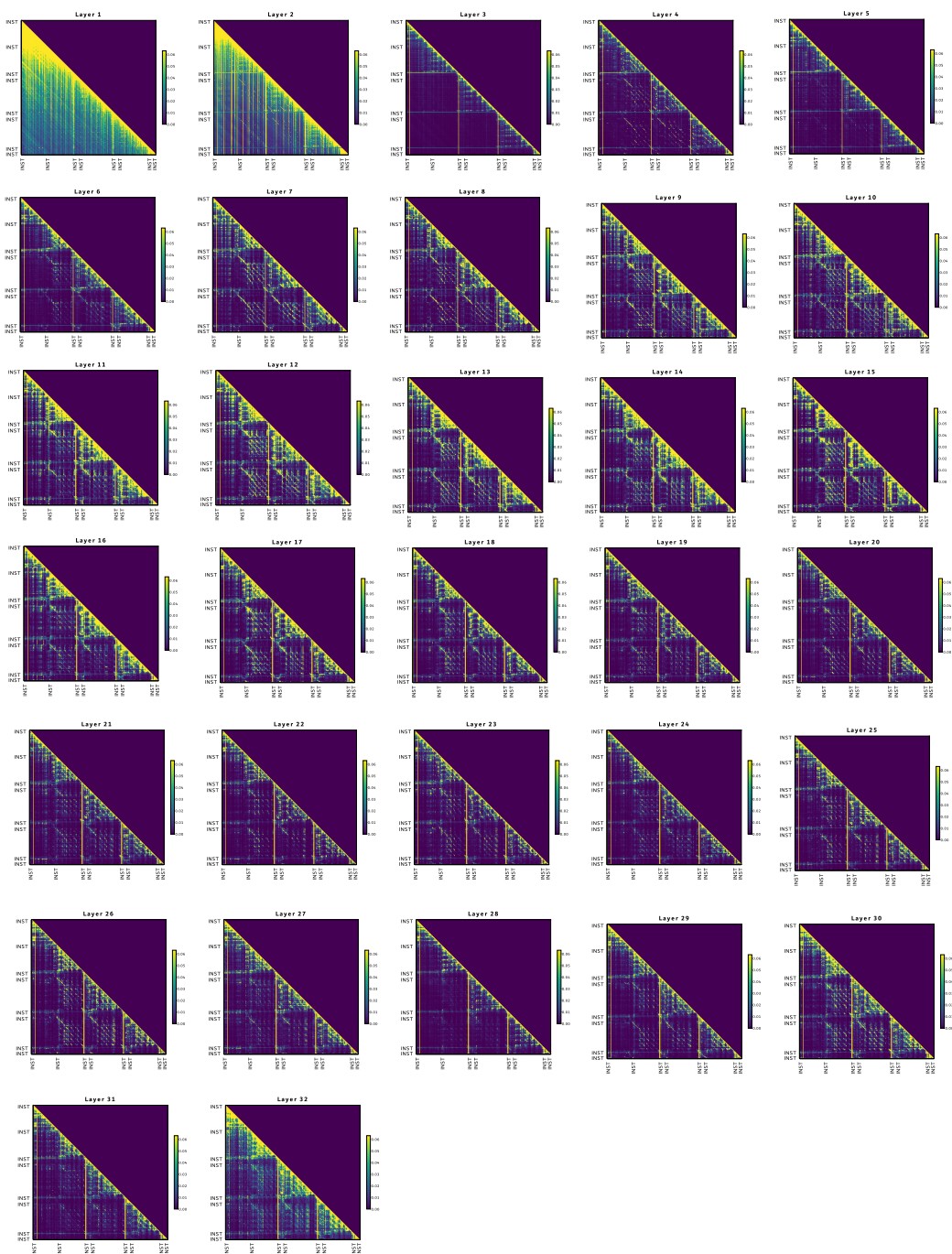

Figure 13: Attention matrices across the hidden layers of the computation of the model $\mathcal{M}$ on the input sequence token $\mathcal{S}$ with given input claim.

# I   Layer-wise Similarities of the Knowledge Graphs

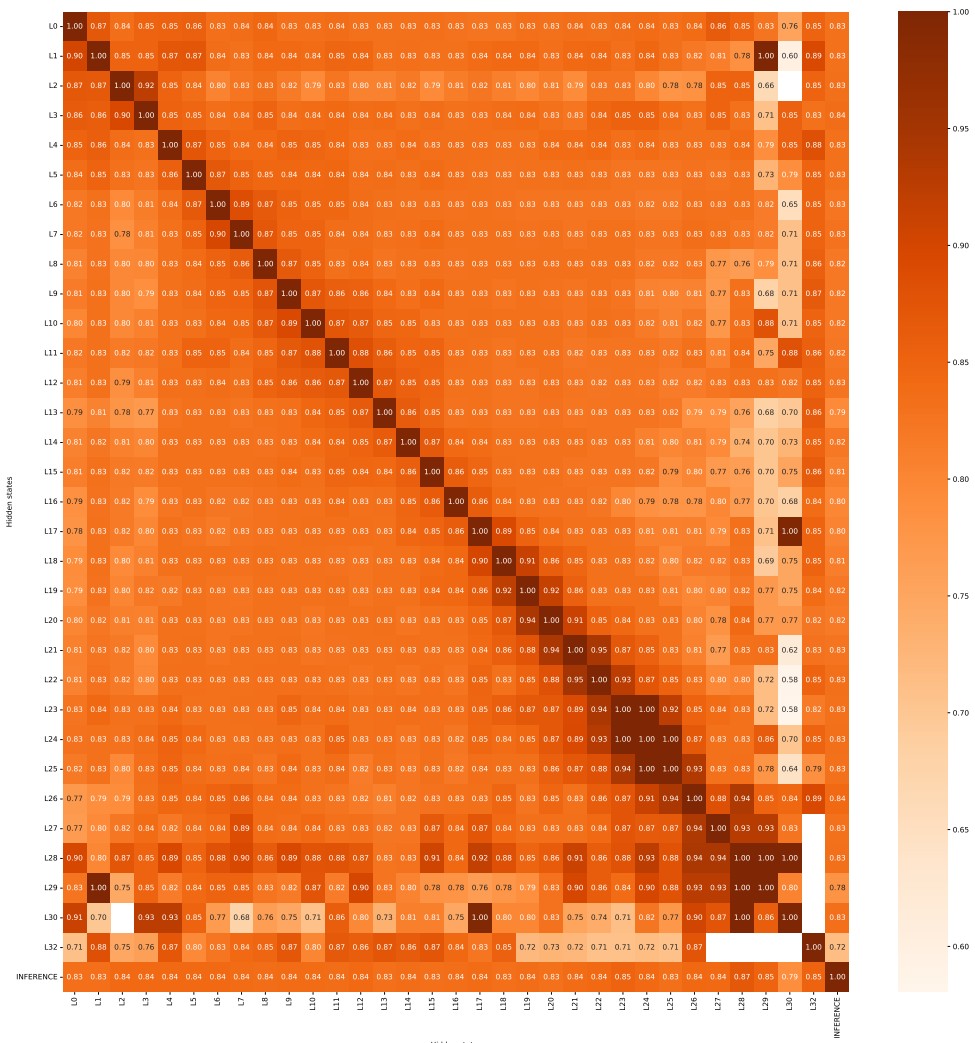

Figure 14: Layer-wise cosine similarities between the graph representation of the factual knowledge decoded from the latent representation of the *l*-th hidden layers.

