# OpenReview forum: "Unveiling LLMs: The Evolution of Latent Representations in a Dynamic Knowledge Graph"
_colmweb.org/COLM/2024/Conference — COLM_

### Official Review · Reviewer_yHaY · 2024-05-09

**Rating:** 7
**Confidence:** 2
**Ethics Flag:** 1

**Summary:**

This paper proposes a framework to understand how Large Language Models (LLMs) recall and use extensive factual knowledge. This paper uses the activation patching method to perform interpretability analysis on two datasets, FEVER and CLIMATE-FEVER. The local interpretability analysis exposes different latent errors from representation to multi-hop reasoning errors. The global analysis uncovered patterns in the underlying evolution of the model's factual knowledge.

**Questions To Authors:**

1. Can the method described in this paper be extended to other tasks involving factual knowledge in Large Language Models (LLMs), especially in AI-assistant dialogue scenarios?

**Reasons To Accept:**

1. This paper analyzes the factual knowledge encoded in the latent representations of Large Language Models (LLMs), conducting a detailed analysis.
2. The framework proposed in this paper appears to have scalability for research on interpretability.

**Reasons To Reject:**

1. The method in this paper is somewhat limited, and how to expand it to a broader range of factual knowledge research is not yet clear.

---

> ### Author Rebuttal · Authors · 2024-05-30
>
> Further applications in factual knowledge research: Section 4.4 showcases one possible use case of our outputs: temporal knowledge graphs for analysing layer-wise similarity of the exploited knowledge in LLM inference, from word-based knowledge to claim-related facts. One could leverage our method, and these graphs, to conduct other graph-related analyses on the underlying process of factual knowledge resolution within LLMs, for example, identify multi-hop reasoning circuits via betweenness, compute the centrality of facts, or study disconnected factual information via communities.
>
> AI-assistant dialogue scenarios: Although our framework focuses on a single inference, applying it to sequential inferences can be an interesting research direction. By expanding its temporal dimension, one could analyse graph overlaps, such as persistent nodes and relationships, among the graph representations of multiple inferences. For example, one could unfold the contextual and factual information the model retains through dialogue, the changes in entity relationships throughout the inferences, or explore the knowledge-related effect of questioning AI’s response. Furthermore, since our probing framework can expose latent errors locally, as visually exposed in Section 4.3, one could evaluate the confidence of the AI assistant’s response by quantifying its internal factuality with a knowledge base at inference time.

---

### Official Review · Reviewer_XVV9 · 2024-05-10

**Rating:** 7
**Confidence:** 4
**Ethics Flag:** 1

**Summary:**

This paper presents a method for probing latent representations of (local) knowledge graphs in an LLM, in a way that is amenable to both local understanding – the representation of a particular statement – and global analysis – looking at how groups of layers behave across examples. The method builds on previous activation patching work from Ghandeharioun et al. 2024 and Zhang and Nanda 2023, but uses more complex prompts to decode full relational triples from the representations of a factual (or false) statement.

The authors validate both their base model inference (M on S) and the patching (M on T) on the FEVER and CLIMATE-FEVER datasets, measuring both standard accuracy as well as self-consistency as a way of validating the decoded facts. They present a variety of interesting qualitative findings, both at the local level - tracing the evolution of a model’s knowledge graph around a single claim - and globally by clustering graphs across layers and identifying common patterns, such as a transition from entity resolution in earlier layers to representing factual knowledge about these entities in the middle layers.

**Questions To Authors:**

- It's not super clear until later in the paper (3.3) what the “temporal” dimension was that the title refers to. Maybe mention in the abstract and introduction that this means over layers of the model?

- I’d be curious to see ablations on the pooling across tokens here. On one hand, it seems like the noun/verb last-token approach gets most of the content words in the input sentence, and so I wonder if it would be much different from just pooling across all tokens. On the other hand, since the model processes left to right, I wonder if it would work to just take the representations of the last token in the claim?

- Did you experiment with any alternative decoding strategies, such as the model to decode the SPO triples directly rather than the relation form? I wonder if these would be easier to generate, as they seem closer to natural language.

- “In the form of ground predicates and without relying on external models or training processes” - I would hesitate a bit to claim there is no training involved, since the method relies on few-shot examples in the prompts. But I don’t think this detracts from the main contributions here.

**Reasons To Accept:**

This paper presents an elegant extension of the methods of Ghandeharioun et al. 2024 to decode full relations, rather than just specific attributes. This enables richer analysis not easily accessible with other probing techniques: an entire (local) knowledge graph can be decoded from the probe output and the evolution of this graph studied over the layers of the model. There are some flaws - see below - but I think this paper presents a unique and novel method that advances the scope of what we can learn about internal representations and layer-wise processing in LLMs.

**Reasons To Reject:**

I have some reservations about the complexity of the measurement technique, as the complex prompts (particularly on the target side) and relatively long-form output leave room for potential bias or skew in the results that may be due to the measurement technique rather than inherent properties of the model. This has been discussed in the literature w.r.t. traditional probing models (e.g. https://arxiv.org/abs/1909.03368), and I worry about the same issues appearing here - for example, the authors discuss at least one case of this where there appears to be leakage from a few-shot example in the target prompt. I would encourage the authors to discuss this more, as this can often be addressed by experimental design - such as including appropriate controls or baselines, marginalizing over different prompts, or corroborating specific findings using simpler methods.

The global analysis also may suffer from this, as the graph node embedding and clustering makes the findings a few more steps removed from the underlying source-side model representations. I wonder if some observations such as the shift from “Robin” to “The Joker” and back to “Robin” (2nd paragraph on page 8) could be artifacts of the discretization steps involved here.

That being said, I think this is at least somewhat orthogonal to the main contributions, and would like to still see this paper published.

---

> ### Author Rebuttal · Authors · 2024-05-30
>
> Technique complexity: Simplifying the target prompt might not significantly affect the findings, such as reducing examples or removing the binary label from the output. On the other hand, training an external probe might be non-trivial, given our desired output: a list of factual information. While the impact of probe complexity is still an open challenge in interpretability, previous studies have probed simpler objects compared to ours such as factual associations (Meng et al., 2022), semantic concepts (Anthropic, 2024) and entity resolution (Ghandeharioun et al., 2024).
>
> Global analysis: It showcases a graph-related use case of our outputs: the temporal KG for analysing layer-wise similarity in factual knowledge. It reveals latent trends similar to other phenomena studied in LLMs, such as word-based focus in early layers.
>
> Shifts among “Robin” and “The Joker”: This insight was directly derived from the textual outputs (see Appendix D), thus, it is not an artefact of the graph-related steps.
>
> Explaining early the temporal concept: We will clarify this aspect in the revised manuscript by introducing it early, as suggested, and using "dynamic" in the title to increase clarity for readers.
>
> Ablations on the token pooling strategy: Using punctuation or the model's special tokens resulted in meaningless output. Next, we tried the input claim’s last token, which provided factual information only about that word. Based on this, we developed a pooling strategy to merge token representations of important input words, considering multiple word representations rather than just the last one. We will include an ablation study in the appendix, showing that adding more tokens, such as stop words, adds noise.
>
> Alternative decoding strategies: We tested SPO triples, finding it slightly less effective in visual comparisons. This representation distracts the model, generating more independent and subject-oriented information, especially with claims with multiple entities (e.g., Empress and Germany).
>
> No training involved: In-context learning, a method of prompt engineering, involves including examples in a model instruction to leverage pre-trained large language models for new tasks without further training. It allows the model to learn by analogy during inference. Conversely, training involves feeding data into a model and adjusting its internal weights through a cost function and backpropagation. We are thus confident that training is not part of our framework.

---

> > ### Comment · Reviewer_XVV9 · 2024-06-07
> >
> > Thank you for your response - I appreciate the clarifications, particularly around token pooling and decoding strategies.
> >
> > > Simplifying the target prompt might not significantly affect the findings, such as reducing examples or removing the binary label from the output. On the other hand, training an external probe might be non-trivial, given our desired output: a list of factual information.
> >
> > I don’t think this necessarily means training an external mlp or linear probe, as you do have a strong argument that those can’t decode the semantic structure you are interested in. But this doesn’t mean you couldn't do some ablations of the measurement technique, for example to study variability / sensitivity to details of the probe design.
> >
> > > Conversely, training involves feeding data into a model and adjusting its internal weights through a cost function and backpropagation.
> >
> > I think your reviewers are aware of this definition :) I did not intend to split hairs on the definition of “training” vs “learning”; but rather that it would not be appropriate to imply that the method does not need at least some amount of labeled data - or particularly, that it’s not sensitive to the way this data (in the prompt) is constructed.

---

### Official Review · Reviewer_GvEw · 2024-05-15

**Rating:** 3
**Confidence:** 5
**Ethics Flag:** 1

**Summary:**

Authors propose an approach to understand the inner workings of Large Language Models. To do so, they chose the task of factual knowledge verification. The idea behind factual knowledge verification is to utilize claims from verification datasets like FEVER and CLIMATE FEVER, converting them into a restricted first-order logic form. They create a prompt and pass it through the model to predict the logical form as well as the label.

**Questions To Authors:**

Suggestion

* Remove the words "Temporal" and "knowledge graph." The term "temporal knowledge graph" is misleading as it triggers the reader to think in a different angle, making it hard to relate to the present approach. Also, the term "temporal" is used loosely, as the layers of LLM; it's better to consider three tiers of layers in LLM: Bottom, Middle, and Top.

**Reasons To Accept:**

Overall, the approach is novel for understanding the internal workings of LLMs.

**Reasons To Reject:**

* Evaluation: While the work is novel, the performance of the proposed approach is nearly similar to or slightly better than a coin toss (50%) since the model's ROC AUC performance is close to 0.5 or 0.6. Given this, it's difficult to determine if the internal workings of LLMs are appropriately evaluated with proposed approach.

* Missing design choices: Throughout the paper, there are several instances where authors have made design choices without explanation. For example, why the study is limited to LLama and not other kinds of models. Is the observation visible only in Decoder-type models or can it also be seen in Encoder-Decoder type models like FLAN? Additionally, what was the intuition behind removing "Not Enough Info"? Moreover, what is the reason to consider only FEVER or CLIMATE FEVER ?

* The approach uses restricted First Order Logic to represent the claim. The language used to represent the claim is restricted as the relations are binary and not n-ary. Hence, they do not accurately represent each of the facts mentioned in the section. The datasets used in FEVER and CLIMATE FEVER are full of sentences containing n-ary relationships among the entities. Therefore, a better representation would benefit the work.

---

> ### Author Rebuttal · Authors · 2024-05-30
>
> We believe there is a misunderstanding about our goal. We do not convert claims into logical forms nor predict them from the model’s internal state. Instead, we unveil the factual information an LLM holds internally when evaluating an input's truthfulness. We will clarify this in the final version.
>
> Evaluation: The ROC AUC figures do not measure the performance of our approach, but simply highlight that internal representations in LLMs do not have enough information to predict the input truthfulness. We will clarify this aspect in the final version.
>
> Limited to LLaMA: We focused on the open-source LLaMA model for its widespread use in interpretability research (e.g., “Language Models Represent Space and Time”, ICLR 2024) and fine-tuning applications (e.g., Alpaca and Vicuna). After submission, we tested other instruction-tuned LLMs (Falcon, LLaMA3, and LLaMA2-13B) and obtained comparable outputs in most cases. These will be added to the appendix.
>
> Consider only two datasets: We chose FEVER as the most popular dataset for factual claims, and CLIMATE-FEVER as its variant including more common-sense claims, which may elicit subjective connections (e.g., "Job killer" vs. "Job creator" entities, Section 4.3).
>
> Excluding claims with not enough info: We considered only claims already verified (supported) or debunked (refuted) through evidence from publicly available information, the FEVER methodology, thus, to avoid prompting the model with unverifiable claims.
>
> Use of restricted First Order Logic: We focused on the popular subject-predicate-object format for compatibility with a knowledge graph representation and the computation of standard similarity metrics over nodes and graphs. The extraction approach can be easily adapted to generate n-ary relations, at the cost of a more complex downstream analysis.
>
> Removing "temporal" and "knowledge graph": We used the term “temporal” to stress the evolution of the model's representations throughout the inference. We will however switch to "dynamic" as Harary and Gupta (1996), to improve the paper’s clarity. On the other hand, we believe the term “knowledge graph” is appropriate to emphasise the structure of our outputs.
>
> Consider three tiers of layers: We generate a distinct graph representation for each hidden layer (Figure 6 and Appendix G). Although our global interpretability analysis, and other studies, reveal similarities in the bottom, middle, and top layers, this dichotomy is observed retrospectively.

---

### Official Review · Reviewer_3hbG · 2024-05-16

**Rating:** 4
**Confidence:** 4
**Ethics Flag:** 1

**Summary:**

**Paper Summary**

This work describes a new method for attempting to provide some interpretability in the context of LLMs when verifying factual claims. More specifically, the work describes a way to construct a temporal knowledge graph, derived from LLM hidden representations, to better visualize working state and reasoning of the LLM during inference. The mechanism uses a few ideas including asking the model to generate a chain of predicates as rationale for claim verification, along with using activation patching to translate hidden representations of different layers of the LLM into knowledge triplets. The work describes mostly qualitative analysis of constructed temporal knowledge graphs when evaluating LLaMA-2 model on the FEVER and CLIMATE-FEVER datasets.

**Review Overview**

Overall, it is not possible to recommend this work for publication in its current form for two main reasons: (1) I have several concerns about the soundness of the techniques employed (that is, the interpretability techniques can be trusted to correctly reflect internal state of the model) and (2) the analysis done using the proposed techniques on FEVER and CLIMATE-FEVER are a bit unclear or redundant given prior work.

In terms of originality, given that this is a heavily explored area, this proposal in this paper adds only a small delta to recent interpretability techniques of LLMs. The “temporal” (layer-wise) aspect of the work is not novel (e.g., https://aclanthology.org/P19-1452/ from 2019 although not explicitly focused on factuality). Augmenting LLM rationales with first-order-logic is also interesting but not quite new (e.g., https://aclanthology.org/2023.findings-emnlp.416/ from EMNLP 2023 and https://openreview.net/forum?id=qFVVBzXxR2V from ICLR 2023).

In terms of clarity of presentation, I found the paper generally clear and well written.

A few potential useful references that are missing:
Dissecting Recall of Factual Associations in Auto-Regressive Language Models - https://aclanthology.org/2023.emnlp-main.751/
https://www.alignmentforum.org/posts/iGuwZTHWb6DFY3sKB/fact-finding-attempting-to-reverse-engineer-factual-recall

**Questions To Authors:**

Besides the comments above, I few more minor questions:

- There are two prompts used in the work (source S, and target T) that are almost the same. I would have expected them to be identical since the patching exercise is attempting to reconstruct the internals of the original inference. Why are these prompts not the same?

- In Section 3.2, the way to construct a summary representation uses some heuristics (averaging, weighting certain terms more or less). Did you try different representations to transfer from inference to patching? What was the impact of those choices? Why not take the entire sequence instead of trying to summarize into a single representation?

- In the results, there is a mention that “ILM” and “Industrial Light and Magic” are different nodes in the knowledge graph. This seems to be just an aliasing problem (that is, the model is likely internally representing the two surface forms almost the same). Why did authors think that this aliasing is causing a multi-hop reasoning problem? How do we know the model internally is not representing these concepts sufficiently similarly?

**Reasons To Accept:**

- The paper is well written and presented making it clear what the problem setting is, the proposed technique and results so far,

- Interpretability of factual knowledge in LLMs, specifically debugging mistakes, is still an open problem and practical tools and results are beneficial to the community.

**Reasons To Reject:**

As mentioned in the summary, the results of the work are still a bit immature. The paper is not thorough in convincing that the techniques used are faithfully representing the internal workings of the LLM and are not, for example, post-hoc explanations.

Specifically:

- One important component of the proposed system is the first-order-logic derivation that the model produces when also producing a verdict of whether a statement is factual. There was never any analysis of whether the logic produced was faithful to the end result. What evidence do we have that the first-order-logic produced is not a hallucination or post-hoc explanation by the model? How faithful is it to its response?

- It is also unclear how faithful the patching mechanism works. It would seem that averaging the output of many tokens (all first-order-logic tokens?) into a single representation would constitute a potential loss of information (as described in Section 3.2). If the first-order-logic contains several entities and relations can we assume that the averaged representation of the tokens can recover the information? Presumably this can be tested in some way (e.g. just for the last/output layer where you know the ground truth first-order-logic and the graph that could be constructed from it),

- The analysis in Section 4 is a bit limited. Aside from general trends in layer-depth representation  (which has been studied previously), there are no new insights derived from the results. It makes me wonder if this may be a reflection of how useful temporal KGs are in practice. How would they be useful to debug specific factuality problems in practice?

---

> ### Author Rebuttal · Authors · 2024-05-30
>
> Faithful of the generated logic: Our aim is not to represent the internal reasoning of the LLM as a sequence of formal logical steps determining the output (which is unlikely given how LLMs are currently trained). Rather, we propose to track the evolution of the factual information as represented at each layer during inference, for which no ground truth is available. Section 4.3 shows that the facts extracted from the internal representations are related to the original input claims.
>
> Layerwise aspect is not novel: existing approaches focus on layerwise representations of single entities on simpler tasks such as entity labelling (Tenney et al., 2019). In contrast, we take a broader perspective by analysing the evolution of knowledge encoded in the layers for entire sentences.
>
> FOL is not novel: While much work has augmented LLM prompts and outputs with FOL, especially in CoT settings, none has analysed LLMs' internal encodings, which operate on vector spaces rather than symbols.
>
> Patching and tokens: We do not average the tokens of first-order logic but sum the vector representations of the input claim’s tokens. This results in a condensed vector representation of the input claim, akin to knowledge distillation, demonstrating an additive property of the encoded semantics.
>
> Graph utility: This structure enhances local interpretability by highlighting entity centrality in LLM reasoning. Globally, graph analyses reveal trends in factual knowledge resolution. Section 4.4 combines well-known phenomena, such as early layers focusing on syntax, with new findings: word-based knowledge evolving into claim-related facts. Additionally, other graph analyses can evaluate fact centrality and disconnected components.
>
> Different prompts: They have distinct purposes. Source is used for the NLP task, eliciting a model behaviour to probe. Target adds more examples to boost in-context abilities during patched inference
>
> Choices on summary representation: Activation patching is a token-to-token and vector-level technique. Without pooling, we cannot inject a multi-token sequence into a token vector. Testing the input claim's final token, we only extract word-based information. We will add an ablation showing that including more tokens, such as stop words, is detrimental.
>
> Aliasing problem: The problem arises when consolidating the company's representation without all the recalled relevant factual information. This might stem from attention mechanisms or catastrophic forgetting.

---

> > ### Author Response · Authors · 2024-06-06
> > **Authors' additions**
> >
> > *Layerwise aspect is not novel*: ... For example, previous studies have analysed simpler objects than factual knowledge such as *factual associations* (Meng et al., 2022 and Geva et al., 2023), *semantic concepts* (Nanda et al., 2023 and Anthropic, 2024) and *entity resolution* (Ghandeharioun et al., 2024).
> >
> > *Patching and prompts*: During the inference on prompt T, the patching technique replaces the vector representation of the placeholder token with the input claim's summary vector representation from the original inference (Figure 4). This patched inference interprets the semantics within this injected vector representation. Essentially, it translates the unknown vector space used internally by the LLM (and altered by merging multiple token representations) into human-understandable semantics. This process is more similar to translation rather than reconstructing the original inference.

---

### Author Response · Authors · 2024-06-05
**Authors' Response to Reviewers' Feedback and Proposed Improvements**

We appreciate the thoughtful and positive comments from all reviewers. You have recognised that our framework is *an elegant extension* of the recent work by Ghandeharioun et al. (ICML 2024, XVV9), *enables richer analysis not easily accessible with other probing techniques* (XVV9); *advances the scope of better understanding internal workings* (GvEw), *representations* (XVV9), *factual knowledge* (3hbG), and *layer-wise processing* (XVV9) in LLMs; provides *scalability for research on interpretability* (yHaY); and the paper is *well-written* and presents the problem setting, the proposed technique and results clearly (3hbG).

We believe that addressing the reviewers’ feedback and questions has significantly enhanced the quality of our manuscript. In the final version, exploiting the additional page, we will improve the following aspects:
### Clarification of the work's objective, the roles of the logic representation and the NLP task
We will provide the following clearer explanation of the objective of our work as well as the purpose of the NLP task and logical representation within our framework (GvEw, 3hbG) in the abstract, introduction, and methodology sections:
- Using latent representations of LLMs, our work uses the NLP task of factual knowledge verification to elicit a model behaviour for probing: exposing the factual information that an LLM holds internally when evaluating the truthfulness of an input; with no ground truth available. We thus neither predict the logical form of claims (GvEw) nor the model outputs (3hbG).
 - On the other hand, we mainly use the logical representation to consistently format the factual information, which supports the evaluation of the input claim, for the subsequent graph generation. Ground predicates, in first-order logic, format the facts related to claims rather than converting the input claims (GvEw). For example, the factual information and the ground predicate “AuthorOf(Hamlet, William Shakespeare)” for the input claim “Edgar Allan Poe wrote Hamlet” (Section 3.1).
### Graph utility
Sections 1 and 4 will better present the advantages of our graph representation, from highlighting entity and fact centrality in LLM reasoning via local analysis to revealing trends in factual knowledge resolution in global one (3hbG, XVV9).
Specifically, we will better introduce the result section by further stressing its showcase purpose, presenting use cases of actual outputs of our framework, rather than as our main contribution. Further works can leverage our framework to conduct other graph-related analyses on LLM’s latent representations.
### Clarification concerning the chosen datasets
Section 4.1 will clarify that the FEVER and CLIMATE-FEVER datasets are used to collect factual and common-sense claims for eliciting a behaviour to analyse (GvEw), as in conventional mechanistic interpretability workflow (Conmy et al., 2023).
These input claims, already verified with publicly available information (GvEw), are exploited to trigger a language model to recall factual knowledge that supports or debunks them.
### Clarification concerning the classification performance
Section 4.2 will better explain the purpose of analysing the classification performance of model inferences. We will rename the section and remove the latent representations’ performance (GvEw) to reduce misunderstandings for readers.
### Confusion with the term “temporal”
- In Sections 1 and 3, we will better introduce and rationalise the temporal dimension of our graph representations. The sequential nature of hidden layers in a transformer-based model allows segmentation within a discrete temporal dimension during inference: layer *t* at time *t*, layer *t+1* at time *t+1*, and so on (GvEw).
- However, based on reviewers' feedback about the confusion and non-conventional use of this term (GvEw, XVV9), we will adopt the more general term “dynamic” in the final version of our manuscript as Harary and Gupt (1996).
### Explanation and ablation concerning the token strategy for the summary representation
We acknowledge the reviewers' questions about the rationality, effects, and alternatives of our strategy for the summary representation (3hbG, XVV9). Section 3.2 of the final manuscript will detail the experimental results that led us to propose such a token pooling strategy:
- We will explain the need for pooling (3hbG) and the unsatisfactory results of using single tokens, such as meaningless texts from punctuation or model’s special tokens, and single-word information for the input’s last token.
- An appendix will include an ablation study with different multi-token strategies, showing outputs when considering all claim tokens or complete words.
### Integration with a preliminary comparison with other models
We will include a visual comparison of outputs generated by other instruction-tuned LLMs in a tabular appendix, testing the latest LLaMA3 (8-billion) as well as LLaMA2 (13-billion) and Falcon (7-billion).

---

### Decision · Program_Chairs · 2024-07-10

**Decision:**

Accept

**Comment:**

The submission builds upon very recent and impactful work in LLM interpretability and presents a novel and useful extension, in particular in that unlike the recent Patchscopes it includes full relations. The negative reviews seem to have several important misunderstandings which are resolvable by a more careful look at the submission and the authors' response. The submission has sufficient amount of empirical results complemented with a qualitative analysis in its current form, and the promised ablation results and clarification improvements are doable and will further improve the quality of the work. I recommend acceptance.